# The Biologist’s Guide to the Glucocorticoid Receptor’s Structure

**DOI:** 10.3390/cells12121636

**Published:** 2023-06-15

**Authors:** Nick Deploey, Laura Van Moortel, Inez Rogatsky, Frank Peelman, Karolien De Bosscher

**Affiliations:** 1VIB Center for Medical Biotechnology, VIB, 9052 Ghent, Belgium; nick.deploey@vib-ugent.be (N.D.); laura.vanmoortel@vib-ugent.be (L.V.M.); frank.peelman@vib-ugent.be (F.P.); 2Department of Biomolecular Medicine, Ghent University, 9000 Ghent, Belgium; 3Translational Nuclear Receptor Research (TNRR) Laboratory, VIB, 9052 Ghent, Belgium; 4Hospital for Special Surgery Research Institute, The David Z. Rosensweig Genomics Center, New York, NY 10021, USA; rogatskyi@hss.edu; 5Graduate Program in Immunology and Microbial Pathogenesis, Weill Cornell Graduate School of Medical Sciences, New York, NY 10065, USA

**Keywords:** glucocorticoid receptor, glucocorticoids, structure, nuclear receptor, N-terminal domain, DNA-binding domain, ligand-binding domain

## Abstract

The glucocorticoid receptor α (GRα) is a member of the nuclear receptor superfamily and functions as a glucocorticoid (GC)-responsive transcription factor. GR can halt inflammation and kill off cancer cells, thus explaining the widespread use of glucocorticoids in the clinic. However, side effects and therapy resistance limit GR’s therapeutic potential, emphasizing the importance of resolving all of GR’s context-specific action mechanisms. Fortunately, the understanding of GR structure, conformation, and stoichiometry in the different GR-controlled biological pathways is now gradually increasing. This information will be crucial to close knowledge gaps on GR function. In this review, we focus on the various domains and mechanisms of action of GR, all from a structural perspective.

## 1. Introduction

Endogenous glucocorticoids (GCs) are steroid hormones synthesized from cholesterol in the zona fasciculata of the adrenal gland cortex. They act mainly through the glucocorticoid receptor α (GRα) but also through the mineralocorticoid receptor (MR), both belonging to the nuclear receptor (NR) superfamily [1]. These steroid hormones are involved in a number of physiological processes including development [2], metabolism [3], the immune response [4], mood and cognitive functions [5,6], cardiovascular function [7,8,9], water and electrolyte balance [10,11] and reproduction [12,13,14].

Cortisol is the endogenous GC in humans, while corticosterone is its rodent counterpart. GCs are essential for life, and their production is tightly regulated by the hypothalamic–pituitary–adrenal (HPA) axis in a rhythmic circadian and ultradian fashion [15,16]. An excess of this hormone results in Cushing’s syndrome, named after the surgeon Harvey Cushing. Symptoms include but are not limited to hyperglycemia, central obesity, striae, thin and fragile skin [17,18]. Depending on whether this excess is caused by factors outside or inside the body, a distinction is made between exogenous and endogenous Cushing’s syndrome (or Cushing’s disease), respectively. The pathology resulting from GC deficiency is termed Addison’s disease (named after the physician Thomas Addison) or adrenal insufficiency. Symptoms include hypoglycemia, weight loss, fatigue and darkened skin [19].

The lipophilic nature of GCs facilitates their diffusion across the cellular membrane, upon which they can induce non-genomic (rapid) and genomic (slower) effects. For the latter, the ligand binds to the GR, upon which the ligand-bound GR translocates from the cytoplasm to the nucleus and regulates gene expression through multiple mechanisms, in a cell- and gene-dependent fashion, as described in [20,21,22,23,24] and further in this review.

Rapid ‘non-genomic’ GC effects (within minutes) have been reported, which do not rely on the function of GR as a transcription factor [25,26]. Non-genomic signaling involves the interference of membrane-bound or cytoplasmic GR with other pathways such as the phosphatidylinositol 3-kinase (PI3K)/Akt and the mitogen-activated protein kinase (MAPK) pathways [27,28]. Additionally, rapid effects can occur through interactions of GCs with the cell membrane (non-specific non-genomic effects) [29]. Here, we focus on the slower (typically hours) and better-characterized GR-mediated genomic effects.

In 1949, the therapeutic potential of compound E (later known as cortisone) was discovered by the physician Philip Hench when treating a patient suffering from rheumatoid arthritis [30]. A year later, the 1950 Nobel Prize in Physiology or Medicine was attributed to Edward Kendall, Tadeus Reichstein and Philip Hench for their discoveries regarding hormones of the adrenal cortex, their structure and their biological effects [31]. Since then, a wide variety of synthetic GCs with altered pharmacodynamics and pharmacokinetics have been synthesized. Examples include dexamethasone, prednisolone and betamethasone. These small molecules are indicated for the treatment of inflammatory and auto-immune diseases such as asthma, inflammatory bowel disease, rheumatoid arthritis, multiple sclerosis and for the treatment of certain types of cancer [32,33]. While short-term systemic GC therapy is generally associated with less severe and reversible side effects, long-term therapy leads to irreversible and more severe side effects, often accompanied by therapy resistance. Nevertheless, both short-term and long-term therapies reduce patients’ quality of life and long-term therapies especially are associated with diminished therapy adherence [34,35].

Amongst the side effects are weight gain, diabetes [36], psychiatric syndromes [37], osteoporosis [38,39], an increased risk of cardiovascular disease [40,41], glaucoma [42] and many more.

## 2. Glucocorticoid Receptor Gene and Isoforms

The human GR is encoded by the nuclear receptor subfamily 3 group C member 1 (*NR3C1*) gene, which is located on the long arm of chromosome 5 (5q31.3) and contains 9 exons (Figure 1). There are fourteen exon 1 splice variants. Four of those exons (1A_1–3_ and 1I) are located within the distal promoter region, while the other ten (1D, 1J, 1E, 1B, 1F, 1G, 1C_1–3_ and H) are positioned within the proximal promoter region [43]. Exon 1 variants are reported to be expressed in a tissue-specific manner and thought to influence GR expression in these tissues [44,45]. The remaining exons 2 through 9 code for the GR protein of which the canonical hGRα-A consists of 777 amino acids (AAs) with a molecular weight of 97 kDa [46,47].

Alternative splicing and alternative translation initiation give rise to a number of GR isoforms (Figure 1). hGRα results from the end of exon 8 being joined to exon 9α, while alternative splicing at exon 9 gives rise to isoform hGRβ. In this isoform, an alternative splice acceptor site joins exon 8 to exon 9β. The resulting proteins are identical up to AA 727, after which hGRα contains 50 additional AAs, whereas hGRβ contains only 15 additional non-homologous AAs [46]. The 742 AA hGRβ with a shorter ligand-binding domain is unable to bind GCs and was originally reported to be constitutively localized in the nucleus. Recent studies, however, revealed that GRβ localization might be cell type-dependent [48]. This receptor variant acts as a dominant-negative inhibitor of GRα but has also been shown to have intrinsic, GRα-independent transcriptional activities [49,50].

When an alternative splice donor site in the intron separating exons 3 and 4 is used, three extra base pairs (GTA) are retained, thereby introducing an Arg in the DNA-binding domain (DBD) between Gly451 and Gln452. This site is near one of the nuclear localization signals of GR, i.e., NLS1, and is located in the lever arm which is part of the DBD. The resulting isoform is referred to as hGRγ (or hGRα-2) (Figure 1). This isoform is more cytoplasmic in the absence of ligand and has a delayed rate of ligand-induced nuclear import compared to hGRα [51,52]. In cell-based assays using glucocorticoid response element-driven (GRE-driven) reporter genes, GRγ exhibited reduced transcriptional activity compared to GRα. On endogenous genes, GRγ activity was generally similar and distinct only at a subset of target genes [53,54]. The GRγ isoform is conserved in mammalian genomes with mRNA expression levels between 5% and 10% of all *NR3C1* transcripts. To date, it remains unclear whether it has a function that is much distinct from GRα [55].

hGR-P diverges from the GRα at the junction of exons 7 and 8, which is separated by intron G, and lacks exon 8 and 9. Due to a failure to splice in this region, a small part of intron G is retained followed by an in-frame stop codon approximately eight base pairs from the 5′ end of intron G. Consequently, a 676 AA isoform is formed, of which the first 674 AAs match hGRα, and a part of the LBD is missing (Figure 1) [56]. hGR-P mRNA levels are increased in myeloma plasma cells, acute lymphocytic leukemia (ALL), non-Hodgkin’s lymphoma (NHL) and, to a lesser extent, in acute myeloid leukemia (AML), compared to normal peripheral blood lymphocytes [57,58].

An alternate splicing event whereby exons 5, 6 and 7 are excised results in the deletion of residues Arg490 to Ser674 encompassing the hinge region and part of the LBD. This results in the 593 AA isoform hGR-A (Figure 1) [56].

hGR-10 (or hGRΔ313-338) arises from a 78 bp deletion in exon 2, resulting in the loss of AAs Gly313 to Thr338 in the N-terminal domain (NTD), between the activation function 1 (AF1) region and the DBD (Figure 1) [59]. The activity of this isoform is largely maintained in luciferase reporter assays, for transcriptional activation and repression [60].

Translation initiation from the first AUG start codon is responsible for the main GRα-A protein product. This first start codon lies within a weak Kozak sequence resulting in alternative translation initiation via leaky ribosomal scanning [61]. Six additional highly conserved AUG start codons exist downstream at positions 27, 86, 90, 98, 316 and 336 and, together with the first AUG start codon, these give rise to seven isoforms with progressively truncated NTDs: hGRα-A, hGRα-B, hGRα-C1, hGRα-C2, hGRα-C3, hGRα-D1, hGRα-D2 and hGRα-D3 (Figure 1) [62].

## 3. Glucocorticoid Receptor Structure

The NR superfamily is a large group of transcription factors divided into seven subfamilies [63]. The GR (*NR3C1*) belongs to subfamily 3 group C, which is also referred to as the 3-ketosteroid receptors. The other 3-ketosteroid receptor members, closely related to GR, are MR (*NR3C2*), the progesterone receptor (PR, *NR3C3)* and the androgen receptor (AR, *NR3C4*) (Figure 2) [63].

With the exception of the atypical receptors DAX (*NR0B1*) and SHP (*NR0B2*), all members of the NR superfamily share a common modular domain organization: A–E (Figure 3) [63].

A/B or the NTD is heterogeneous in size, the least conserved among the different NRs and mostly disordered in solution. This domain contains the majority of the known post-translational modification (PTM) sites and contains the AF1 region, which is important for the interaction with different coregulators [63].

C or the DBD is smaller than the NTD in the case of the 3-ketosteroid receptors and is the most conserved domain among the NRs. Two zinc finger motifs in this domain determine DNA binding specificity and receptor dimerization [63].

D or the hinge region is poorly conserved in size and sequence among the NRs. It is a flexible linker between the DBD and LBD which is also prone to PTMs. This region can contain a nuclear localization signal (NLS) [63].

E or the LBD is a structured domain and contains a hydrophobic ligand-binding pocket (LBP). Variation in the LBP determines the NR ligand-binding specificity. Ligand binding leads to structural stabilization of the second region for the interaction with coregulators, termed activation function 2 (AF2) [63].

### 3.1. N-Terminal Domain (NTD)

As stated above, the NTD is the least-conserved domain amongst the NR family, hence also the least-conserved domain in GR. A secondary structure prediction revealed a large portion as having a random coil configuration and a small portion as having a helix and sheet structure [64]. This is unlike the DBD and LBD, where significantly more secondary structure elements are found [64]. No single NR NTD crystal structure has been determined, owing to this domain’s intrinsically disordered nature.

#### 3.1.1. Secondary Structure

In hGRα, the NTD encompasses AAs 1 to 421 and makes up the largest part of the receptor. A subregion within this large domain termed the activation function 1 (AF1), also called tau1 (τ1), spans residues 77 to 262. This region acts as a hub for the interaction with certain coregulatory proteins and is therefore required for full transcriptional activity of GR [65,66,67,68]. Consequently, isoforms hGRα-D1, -D2 and -D3, which lack AF1, display diminished transcriptional activity [62]. Additionally, coregulatory proteins differentially interact with the AF1 and AF2 (in the LBD) regions [69]. A 41-AA core region (τ1_C_ or AF1_C_) within τ1 (hGRα residues 187–244) was identified as the most important part for transcriptional activity [70]. In an aqueous solution and at variable pH, this τ1_C_ region was mostly unstructured [71]. In the presence of a secondary structure-promoting agent, trifluoroethanol (TFE), the τ1_C_ region acquired three regions with alpha-helical character (helix 1: 189–201; helix 2: 216–226; helix 3: 234–239). The disruption of these helical structures through helix-breaking Pro substitutions reduced GR’s transcriptional activity [71]. By making combinations of these helical regions, it was found that a single helical segment is incapable of establishing transcriptional activity, whereas any combination of two or three segments is sufficient [72]. A more recent study [73] investigated whether τ1_C_ contains transient secondary structures, also called Pre-Structured Motifs (PreSMos [74]), in aqueous solution. They found three helical PreSMos, in concordance with [71], but at slightly shifted positions (helix 1′: 185–202; helix 2′: 206–225; helix 3′: 232–244) [73].

A thermodynamics study using the protective osmolyte trimethylamine N-oxide (TMAO) to induce folding transitions in the NTD of hGRα-A, and two of its translational isoforms (hGRα-C2 and hGRα-C3) identified at least two thermodynamically coupled regions [75]. The first was a functional region containing the AF1 site, the second a regulatory region (hGRα residues 1–97) acting as an inhibitory domain. Shortening the regulatory region as seen in the translational isoforms (Figure 1) increased the stability and activity of the receptor. Thus, the most truncated isoform included in the study, hGRα-C3, showed the highest stability and doubled activity, followed by hGRα-C2 and, lastly, hGRα-A [62,75].

#### 3.1.2. Phosphorylation

GR contains at least seven experimentally confirmed phosphorylation sites conserved among humans, rats and mice. In hGR, these sites are located at Ser113, Ser134, Ser141, Ser203, Ser211, Ser226 and Ser404 (See Figure 4 for cross-species sequence alignment) [22,76,77,78,79,80,81]. Remarkably, all sites are situated within the intrinsically disordered NTD, and it has been shown that phosphorylation at Ser211 induces a functionally active folded conformation of tau1c. The phosphorylation-deficient S211A mutant did not show any significant structural rearrangements, whereas the S211E mutation (to mimic the phosphate group negative charge) only moderately increased helical content. Additionally, Ser211 phosphorylation of AF1 led to a significantly increased interaction with the coregulators TATA-box binding protein (TBP), CREB binding protein (CBP) and nuclear receptor coactivator 1 (NCOA1 or SRC-1) [78].

Luciferase reporter assays using GR500, a receptor lacking the LBD and transcriptionally active in the absence of ligand, revealed that GR500-S211A lost 75% of its transcriptional activity compared to wild-type GR500, which was only partially recovered by GR500-S211E. The activity of GR500-S203A was reduced by ~50%, and this was mostly recovered by GR500-S203E. Similar results were observed for GR500-S226A and GR500-S226E, respectively [82]. The double-mutant GR500-S203A/S226A showed an approximately 50% decrease in transcriptional activity, comparable to the GR500-S203A and GR500-S226A single mutants. On the other hand, when GR500 contained the S211A mutation in combination with either S203A or S226A, most of its transcriptional activity was lost. Similarly, nearly all activity was lost in the GR500-S203A/S211A/S226A triple mutant [82]. In conclusion, these data indicate that SER211 phosphorylation is most crucial for the transcriptional activity of GR500.

When the AF1_C_ peptide was phosphorylated at Ser203, Ser211 or Ser226, unique structural rearrangements were observed in silico via energy minimization [82]. Hydrogen bonds were detected between P-Ser203 and Lys206 and also between P-Ser211 and Arg214 [82]. Experiments with individually phosphorylated AF1_C_ peptides suggested that P-Ser203 and P-Ser211 prefer to interact with Lys206, whereas P-Ser226 makes a weak contact with Arg214 [82]. Furthermore, it appears that P-Ser203 and P-Ser226 induce local structural rearrangements, while P-Ser211 promotes both local structural rearrangements as well as an overall more compact structure in the AF1_C_ peptide [82]. It is proposed that the region surrounding Ser211 is critical in GR transcriptional activation, and this is supported by the reduced transcriptional activity observed upon mutating the close-by Trp213 [82,83].

In line with the importance of P-Ser211 reported above, phosphorylation at this site was found to be a more reliable predictor of GR ligand effects on endogenous GR target genes than luciferase reporters evaluating GR-driven gene activation and gene repression [84]. This advocates for the inclusion of this important parameter when assessing the effect of ligand on gene regulation and when screening for ligands with improved therapeutic benefit [84].

#### 3.1.3. Intrinsic Disorder of the NTD

It has long been hypothesized that the intrinsically disordered nature of NTD enables it to acquire distinct structures upon binding different cofactors. Indeed, a folded functional conformation can be induced in the NTD in vitro by the osmolyte TMAO, and this conformation selectively binds TBP, CBP and NCOA1 [68]. AF1 can also acquire structure through binding events outside of this region, as was demonstrated in early studies using binding of a two-domain GR, lacking the LBD, to a palindromic glucocorticoid response element (GRE) [85]. Additionally, the interaction of TBP and nuclear receptor coactivator 2 (NCOA2, also known as SRC-2, TIF2 or GRIP1) to GR AF1 induced a significant amount of helical structure in the latter, suggesting that coregulator binding increases the amount of structure in AF1 [86,87,88]. Lastly, accumulating evidence shows that proteins use their intrinsically disordered regions (IDRs) to form phase-separated condensates to drive the transcription of genes [89,90,91,92]. Phase separation is a process by which molecules in a solution or mixture spontaneously separate into two or more distinct phases, such as liquid droplets or solid aggregates. It has been shown that GR forms such condensates both in vitro and in vivo [91,92]. Single-molecule analysis of the GR in the nuclei of murine mammary 3617 adenocarcinoma cells revealed two distinct states of reduced mobility [92]. The first state accounts for specific binding to chromatin and is DBD-dependent [92]. The second state is an IDR-dependent ‘confinement state’, whereby the localization of GR is restricted to a confined area in the nucleus by means of phase separation, promoting binding events that are highly specific [92].

### 3.2. DNA-Binding Domain (DBD)

#### 3.2.1. DBD Structure

The DBD is located C-terminally from the NTD and encompasses residues 421 to 486 in the hGR. The rat homologue of this domain in complex with DNA was the first to be structurally resolved in 1991 (see Table 1 for an overview of all GR DBD crystal structures) [93]. The DBD can be divided into two highly conserved zinc finger subdomains, separated by a flexible lever arm (Figure 5). Each zinc finger comprises four cysteine residues coordinating a single Zn^2+^ ion, an amphipathic helix and peptide loop [93]. The first subdomain contains the proximal box (P-box; residues 439–443), required for DNA-specific contacts. Helix 1 of the first subdomain (H1, residues 438–451) is positioned in the DNA major groove and assists in DNA-binding via base-specific contacts. For this reason, H1 is often referred to as the DNA-recognition helix. The lever arm (residues 450–456) is a flexible loop whose conformation is allosterically regulated by the sequence of the bound DNA, yielding different transcriptional responses [54,94]. The second subdomain contains the distal-loop (D-loop or D-box, residues 458–462) involved in GR dimerization [95]. A distorted helix 2 (H2, residues 468–472) in the second subdomain was described by Luisi et al. but has not been unambiguously observed in NMR experiments [93,96,97]. Recently, however, Frank et al. reported the same conformation for the distorted helix in DNA-bound and DNA-free GR DBD (PDB ID: 3G99 and 6CFN, respectively), both resolved using X-ray diffraction [54,98]. Lastly, helix 3 (H3, residues 473–485) from the DBD and helix 4 (H4, residues 489–495), located in the hinge region, make contact with the DNA minor groove [54,94].

The GR is able to activate and repress its target genes through various mechanisms, either by direct binding to DNA or by binding to and modulating the function of other DNA-bound transcription factors (tethering). The best-described mechanism is gene activation through binding to a canonical GRE [93]. Similarly, the GR can also bind to inverted-repeat GREs (IR-GREs) for the repression of genes [100,106]. Binding of monomeric GR to a canonical half-site with consensus sequence AGAACA (or the inverse complement TGTTCT) has also been reported and can lead to both up- and downregulation of target genes [107,108]. In specific cases, GR has been shown to repress the pro-inflammatory transcription factors activator protein 1 (AP-1) and nuclear factor-κB (NF-κB) by binding to DNA sequences interspersed with binding sites for those factors [102,104]. For the former, GR binds directly to a GRE-like half-site located within a canonical AP-1 TRE (TGA(G/C)TC) [104]. For the latter, GR binds a cryptic response element (AATTY, Y = pyrimidine base) between the binding footprints of NF-κB subunits within κBREs [102]. In addition, the GR DBD is capable of binding biological and synthetic RNAs of which Gas5 is the most thoroughly researched [109,110,111,112,113]. The final mechanism, referred to as tethering, does not involve direct DNA contacts but is mediated by various protein–protein interactions [114,115,116,117,118].

#### 3.2.2. GR Binding to Glucocorticoid Response Elements (GREs)

NR DBDs bind to specific DNA sequences termed nuclear receptor response elements (NREs), and small differences between RE sequences can guide receptor specificity [119]. The canonical GRE is 15 base pairs long and composed of two hexameric inverted-repeat half-sites, separated by a three-base-pair spacer. Its consensus sequence is 5′-A_1_GAACAnnnTGTTCT_15_-3′ [120]. GREs from the same gene were shown to be highly conserved between four mammalian species (human, mouse, dog and rat), but GREs from different genes show distinctive differences in 10 out of 15 base pairs [121]. However, six GRE bases are contacted directly by the GR dimer and show a significant nucleotide preference (Figure 6, underlined positions) [95]. Remarkably, nucleotides that are not directly contacted by GR also show pronounced nucleotide preference, as demonstrated for the spacer nucleotides at positions 7–9, which are usually pyrimidines (C and T). Depending on the GRE context, changing spacer sequence can impact transcriptional activity [95]. The nucleotides adjacent to the spacer, at positions 6 and 10, are preferably A and T, respectively [95]. Almost no base preference was identified for positions 3 and 13 [95].

Additionally, reversing the sequence of asymmetric GREs from Sgk and GILZ relative to the transcription start site (TSS) negatively impacted transcriptional activation but not affinity for the GR DBD [54]. In addition, luciferase reporter assays with GREs derived from endogenous target genes displayed significant differences in activity when their first hexameric half-site (AGAACA) was identical but their spacer and second half-site (XGTYCN, X = T/G, Y = A/T/C, N = any nucleotide) varied [54]. A later study involving GREs that differ only at nonspecific bases of the spacer or at half-site positions 13 and 15 concluded that the spacer sequence, within the context of the whole GRE, does influence GR activity [95].

In a study from the Yamamoto team in 2009, rat GR DBD was found to bind the GRE as a head-to-head dimer with their D-loops facing each other, whereby the first monomer contacts the first hexameric half-site, and the second monomer contacts the second half-site (Figure 7) [54]. The head-to-head arrangement was also observed in the recent hGR multidomain crystal structure [122]. The recognition helix of the first monomer involving rGR Lys461, Val462 and Arg466 (see Figure 4 for cross-species sequence alignment) makes specific major groove contacts [54,113]. This is similar for the second monomer but with a stronger Lys461 contact and lacking the Val462 contact [54,113]. Residues 509–515 form the C-terminal helix 4 (H4, residues hGR 489–495, mGR 506–512) which lies across the minor groove [54]. A non-specific contact between the minor groove backbone 3 bp upstream of the GRE and H4 is mediated by Arg510 (hGR Arg491, mGR Arg507). Mutating this residue to an alanine (R510A) reduced the affinity for DNA by ~three-fold yet increased transcriptional activation [54]. Several crystal structures revealed that differences in GRE, including those that do not directly contact GR, conferred changes in the lever arm [54]. Notably, residue His472 (hGR His453, mGR His469) in the lever arm adopted distinct conformations. In the first GR monomer contacting the first invariable half-site (AGAACA), His472 was packed in the core of the protein fold, whereas in the lever arm of the second monomer, contacting the second variable half-site (XGTYCN, X = T/G, Y = A/T/C, N = any nucleotide), it was flipped out (Figure 7). Additionally, the conformation of the lever arm in the second monomer was more variable across structures with different GREs, especially when spacer length varied [54]. A comparison of the Pal-F (spacer = AAA) and Fkbp5 (spacer = GGG) GREs revealed a more narrow minor groove for the former [95]. It was proposed that the width of the minor groove imposes structural constraints on lysine- and specifically Lys490 (hGR Lys471, mGR Lys487)-mediated backbone contacts with DNA [95]. [^1^H,^15^N]-heteronuclear single-quantum coherence spectroscopy (^15^N-HSQC) was employed to identify which regions of the DBD are affected by differences in spacer sequence vs. those in half-site positions 13 and 15 [95]. Changes in the spacer sequence affected Ala477 (hGR Ala458, mGR Ala474), Gly478 (hGR Gly459, mGR Gly475), the DNA-recognition helix (H1) and the lever arm, whereas changes at positions 13 and 15 influenced outward-facing surfaces of the DBD near the DNA, as well as the DNA-recognition helix and lever arm [54,95]. Importantly, changes in the DNA-binding interface are communicated to the lever arm and to the other dimer partner via the D-loop [95]. On top of this, an NMR study of the GR LBD revealed allosteric communication between the LBP and the N-terminal end of DBD H1, implying inter-domain communication [123]. Consistently, the first GR multidomain (DBD and LBD) crystal structure described an interface between the H1 from the LBD of the first monomer (LBD1) and the DBD D-loops of both monomers, suggesting a path for allosteric communication between the bound DNA sequence and the ligand within the LBP [122].

GRγ represents a pre-eminent isoform to study the lever arm due to an arginine insertion in this region. Relative to the GRα, this isoform showed reduced activation in GRE-dependent reporter assays [53,54] yet almost equal repression of the osteocalcin promoter-derived reporter [124]. The regulation of endogenous GR targets by GRγ was mostly similar to that by GRα with a few exceptions [54]. Again, DNA-bound GRα and GRγ crystal structures were similar except for the lever arm [54]. In the crystal structure of hGRγ, it appears that R471 forms a weak hydrogen bond with the DNA backbone at one of the half-sites, which can explain the relatively higher affinity of GRγ for DNA compared to GRα [125]. Changes in the DNA-recognition helix (H1) uncovered by NMR studies support this idea [125]. Lastly, it is remarkable that the difference in binding affinity between GRα and GRγ is smallest for the Pal GRE (with the narrowest spacer) and largest for the FKBP5 GRE (with the widest spacer). A possible explanation could be that the introduction of a residue in the lever arm might relieve strain imposed on the dimer interface spanning the spacer [125].

Seven additional rGR lever arm mutants analyzed (E469A, G470A, Q471A, H472A, H472R, N473A and Y474A; see Figure 4 for cross-species sequence alignment) produced GRE-specific transcriptional effects suggesting that subtle differences in the lever arm can indeed yield quite distinct GR activities [54].

The GR monomer–dimer paradigm, in which monomer-driven anti-inflammatory effects of GR were believed to be separable from the dimer-driven side effects, originates from experiments involving GR DBD mutants. In 1994, the importance of hGR Ala458 was experimentally tested by mice carrying a substitution in this D-loop, termed GR_dim_ [126]. The D-loop is involved in GR dimerization due to reciprocal hydrogen bonds between the carbonyl of rGR Ala477 (hGR Ala458, mGR Ala474) of one monomer and the amide of Ile483 (hGR Ile464, mGR Ile480) of the other [95]. The activity of the GR_dim_ mutant in GRE-driven reporter assays is reported to be lower, equal or higher than the WT depending on the GRE [95,127,128]. This mutant was also shown to have reduced affinity for DNA, which can be attributed to diminished cooperativity between the mutant’s monomers. The rGR A477T mutation causes local structural changes in the D-loop and residues surrounding Ile483 as well as reorganizations in the N-terminal part of the lever arm and the DNA recognition helix [95]. Later on, it was shown that GR_dim_ is still able to dimerize [128], suggesting the existence of an additional dimerization interface. In line herewith, dimerization is heavily impaired for the double-mutant hGR A458T/I628A (GR_mon_) [129].

#### 3.2.3. GR Binding to Inverted-Repeat Negative Glucocorticoid Response Element (IR-nGRE)

Agonist-bound GR, but not antagonist-bound GR, is also able to repress genes through direct DNA binding at so-called inverted-repeat negative GREs (IR nGREs) and subsequent assembly of corepressor complexes which consist of SMRT/NCOR corepressors and histone deacetylases (HDACs) [106,130]. The GR is the only 3-ketosteroid receptor able to bind the *TSLP* IR-nGRE with nanomolar affinity, in vitro [101]. Three key substitutions in GR lineage (I420L, G425S and F478Y) interacted epistatically and enhanced repression at IR-nGREs [101]. The consensus IR-nGRE sequence was defined as 5′-CTCC (N)0–2 GGAGA-3′ [100,106]. In the thymic stromal lymphopoietin (*TSLP*) gene, the inverted-repeat motifs are separated by 1 bp, but in vitro work revealed that the lack of a spacer or a 2 bp spacer was also tolerated [106]. Structural analysis revealed that two GR monomers bind to nonidentical everted half-sites, a high-affinity and a low-affinity half-site, in a head-to-tail fashion and on opposite sides of the DNA, with their D-loops facing away from each other (Figure 8) [100]. The high-affinity half-site maintains relatively constant binding affinity between different IR-nGREs, whereas the low-affinity half-site displays substantial variation amongst IR-nGREs [100].

GR binding to the *TSLP* IR-nGRE (5′-AGC_-1_T_0_ C_1_T_2_CC_4_ G GGAG_9_G_10_C_11_G-3′) revealed that the GR monomer bound to the high-affinity half-site made three base-specific contacts in the DNA major groove (Figure 8) [100]. Val443 is responsible for two hydrophobic contacts with C_1_ and T_2_ [100]. Lys442 donates a hydrogen bond to N7 of G_4_ of the opposite strand and mutating this guanine (G_4_) to adenine attenuates binding of GR [100]. Similarly, mutating Lys442 to Ala is detrimental for IR-nGRE binding [100]. Arg447 makes base-specific contacts on a positive GRE, which is impossible on the IR-nGRE due to a steric clash with T_0_. Instead, Arg447 makes hydrophobic interactions with T_0_ and ionic interactions with the C_-1_ backbone phosphate. Mutating T_0_ to guanine enables Arg447 to make base-specific contacts, but repression functionality is lost [106].

The low-affinity half-site only involves an Arg447 residue making a sequence-specific contact with G_11_ on the opposite strand and outside the IR-nGRE consensus sequence (Figure 8) [100]. Mutating G_11_ did not affect the GR monomer binding to the high-affinity site. Lys442 and Val443 from the monomer binding to the low-affinity half-site do not reach far enough into the DNA major groove to make base-specific contacts [100]. The low-affinity half-site is far more resistant to mutations, and its function remains unclear [100].

Binding of the first GR monomer to a GRE narrows the minor groove thereby facilitating dimerization. In contrast, GR binding to IR-nGRE widens the minor groove and narrow the major groove [100]. Moreover, the dimerization loops of both GR monomers are rotated 180° from the DNA axis facing away from each other [100]. Together, these changes result in the strong negative binding cooperativity observed.

A major role was ascribed to the lever arm for translating GRE sequence differences into GR structural differences [54]. For GRE-bound GR, residue His472 (hGR His453, mGR His469) of the lever arm adopted a packed configuration for the first monomer and a flipped-out configuration for the second monomer [54]. At IR-nGRE, this configuration was flipped for both monomers and stabilized by hGR Arg447 and Tyr455 (Figure 8) [100].

The dimerization-compromising mutant hGR A458T (GR_dim_) binds to GREs less cooperatively compared to wild-type GR [100]. At IR-nGRE sites, binding to the low-affinity site was increased, and binding to the high-affinity site was decreased with an overall three-fold reduction in binding affinity [100]. Consequently, GR_dim_ exhibited slightly lower repressive capabilities compared to wild-type GR [100].

#### 3.2.4. GR Binding to Half-Sites

The α-amylase 2 gene and the CYP3A5 member of the cytochrome P450 gene family are examples of genes harboring just two isolated GR half-sites in their regulatory region and which are crucial for gene expression [131,132]. A survey of GREs bound by rGR_dim_, revealed a half-site motif and suggested that rGR_dim_ binds as a monomer in vivo and is able to regulate many genes. The subsequent analysis of rGRα GREs suggested two modes of binding: the first to consensus half-sites (AGAACA), most likely bound by monomers, and the second to degenerate full sites, most likely bound by dimers [107]. In mice, GR ChIP-exo on liver after endogenous corticosterone exposure revealed that monomeric GR binding to a half-site motif is more frequent than homodimer binding to canonical GREs and that the former also drives transcription [108]. Treatment with exogenous GCs (prednisolone), led to an increase in GR dimers at ligand-activated genes and a decrease in GR at half-site motifs [108]. At the time of writing, the Protein Data Bank contains three PDB IDs (6BQU, 6BSE, and 6BSF) for GR binding to a half-site [133]. However, the corresponding literature describing the work has not yet been published.

#### 3.2.5. GR Binding to TRE

The activator protein-1 (AP-1) proteins are an extensive family of dimeric leucine zipper transcription factors that bind to 12-O-tetradecanoylphorbol-13-acetate (TPA) responsive elements (TREs) and are known to activate a wide variety of genes, including many encoding inflammatory cytokines and chemokines [134]. The GR was shown to directly bind particular TREs through an embedded GRE-like half-site and consequently repress transcription at these sites [104].

The crystal structures of GR DBD bound to the IL11 (T_1_GACTC_6_) and the VCAM1 (T_1_GAGTC_6_) TREs were determined (Figure 9 and Figure 10, respectively) (see also Table 1) [104]. In both structures, the first GR monomer recognizes a hexameric TGA(G/C)TC of which the degenerate fourth base is not directly contacted by GR. In both the GR DBD:IL11 and the GR DBD:VCAM1 TRE complexes, the DNA-reading helix is positioned in the major groove; however, three side chains (Val443, Lys442 and Arg447) make base-specific contacts in the former complex, whereas only two side chains (Val443 and Arg447) are involved in base-specific contact in the latter complex. In both, Arg447 makes a hydrogen bond to the N7 position on G_2_ and van der Waals contacts to the methyl on T_1_, while Val443 makes van der Waals contacts to G_2_. In the GR DBD:IL11 TRE complex, Lys442 makes additional hydrogen bonds to the N7 position on A_5_ on the opposite strand. This base-specific interaction is not observed in the GR DBD:VCAM1 TRE complex, where GR makes contacts with the backbone of A_5_ on the opposite strand instead. The second GR DBD monomer was bound to the opposite side of the TRE DNA in an everted fashion and does not make base-specific contacts. It is likely that this GR DBD monomer is important for efficient crystal packing but unlikely to be relevant in vivo, as corroborated by the cell-based reporter and NMR footprinting assays [104].

#### 3.2.6. GR Binding to κBRE

Nuclear factor-κB (NF-κB) encompasses the Rel family of dimeric transcription factors, which are central effectors of the inflammatory gene expression program [135]. NF-κB subunits bind to specific DNA sequences termed κB response elements (κBRE) and activate transcription of numerous pro-inflammatory genes. GR represses inflammatory gene expression by NF-κB through a variety of mechanisms, with GR tethering to DNA-bound p65/RelA as the most well-described mechanism. However, it was recently reported that in some cases, GR can recognize an evolutionary conserved cryptic response element located between the binding sites of the NF-κB subunits [102].

The crystal structures of GR DBDs bound to κBREs from five genes, CCL2, ICAM1, IL8, PLAU and RELB, were solved (Figure 11 and Table 1). The study describes a GR dimer in which the second monomer is unlikely to be relevant in vivo, analogous to GR binding to TRE. In each structure, GR DBD recognized an A_1_ATTY_5_ (Y = pyrimidine base) sequence through contacts mediated by Lys442, Val443 and Arg447. Lys442 makes a hydrogen bond to the purine residue on the opposite strand of the terminal base, Y_5_. Val443 makes van der Waals contacts with T_3_ on the opposite strand, and their distances are rather constant across all structures. Finally, the guanidino group of Arg447 makes van der Waals contacts with adenine A_1_, whereas its terminal amine forms a hydrogen bond with A_2_ [102].

The identification of novel DNA-binding-dependent mechanisms by which the GR can repress pro-inflammatory genes may have implications for the strategies of how to search for improved GR therapies [102,104]. What the determinants are that drive the selection between tethering and DNA-binding-dependent repression and what the contribution is hereof in vivo has yet to be uncovered.

#### 3.2.7. GR Binding to RNA

GR is reported to bind several RNAs including transfer RNA (tRNA), messenger RNA (mRNA) and growth-arrest-specific 5 long noncoding RNA (Gas5 lncRNA) [109,110,111,112,136].

Ten small nucleolar RNAs (snoRNAs) and two spliced isoforms, Gas5a and Gas5b, are expressed from the *gas5* gene [109]. Gas5 lncRNA accumulates upon cellular growth arrest (hence the name) and consequently contributes to cell death via its pro-apoptotic effects [137,138,139]. Gas5a lncRNA was found to associate with GR in an RNA and protein co-immunoprecipitation assay in HeLa cells, and this interaction was GR DBD dependent [109]. Furthermore, Gas5a lncRNA inhibited dexamethasone-induced GR transcriptional activity in reporter assays [109]. The effects of Gas5a lncRNA overexpression were also tested on several GC-responsive genes harboring endogenous GREs, including glucose-6-phosphatase (G6Pase or G6PC1), glucocorticoid-induced leucine zipper (GILZ or TSC22D3), phosphoenolpyruvate carboxykinase 1 (PEPCK-C or PCK1) and serum/glucocorticoid-regulated kinase 1 (Sgk1). Gas5a lncRNA reduced mRNA expression of these genes and GR occupancy at their GREs in a dose-dependent matter [109]. This reduction in GR DNA-binding by Gas5 has pro-apoptotic consequences, and consequently, downregulation of Gas5 was shown to have anti-apoptotic effects in cell culture [110]. Consistently, in prostate and breast cancers, the downregulation of Gas5 is correlated with poor prognosis [140,141].

Gas5a and Gas5b lncRNA are 598 and 630 bases long, respectively, and form numerous hairpin structures [109]. The GR DBD binds the consensus GRE and Gas5 with comparable affinities in the 67–125 nM range [100,112]. Although GR binds GREs as a dimer with slightly positive cooperativity (Hill coefficient of 1.3–1.4), this is not the case for GR-RNA binding (Hill coefficient of 0,93) [100,112,113]. NMR spectra revealed that the GR dimerization loop was not affected by binding to a 33-nucleotide Gas5 RNA hairpin. This suggests that GR binding to Gas5 is dimerization independent [110]. Furthermore, the GR A458Tdim mutant shows a ~three-fold weaker binding affinity for GREs compared to wild-type GR, while maintaining full affinity for Gas5, providing additional evidence to support this conclusion [112]. Earlier reports suggested that the region between nucleotides 400 and 598 in Gas5 is responsible for binding GR as this region contains a GRE mimic (Gas5 GREM), and mutations in GREM compromised GR binding [109,110]. However, these earlier studies utilized Gas5 constructs lacking an RNA terminal loop. A more recent study found that mutations in the GREM had no effects on GR binding and proposed instead that GR binds in a structure-specific rather than sequence-specific fashion [112]. This was supported by in vitro and in silico observations showing that a loss of the Gas5 RNA terminal loop resulted in a drastic decrease in GR binding [112,113]. A 4 bp stem-loop proved sufficient for binding, and increasing the loop length or altering its sequence had no significant impact on binding [112]. However, a 3 bp stem-loop did display reduced affinity for GR [110]. In line with these findings, GR was found to bind to multiple biological and synthetic RNA hairpins that had no sequence similarity to the Gas5 GREM, which implies that many RNAs have the potential to impact GR biology [112].

The DBD shares the highest percent identity among the steroid hormone receptors (Figure 2). Consequently, it is no surprise that Gas5a lncRNA was shown to bind the DBD of the AR, MR and PR-A and to suppress their transcriptional activity in a ligand-dependent fashion, in vitro [109,110]. The DBD of ERα shares the least sequence identity with the GR DBD and accordingly, Gas5a lncRNA does not bind to the ERα DBD in vitro [109]. The ERα Glu203 is considered critical for DNA binding as it contacts a cytosine in the ERE. In contrast, the residue at this position in other steroid receptors is a Gly and does not contact the DNA. Therefore, the bulkier Glu203 in ERα prevents binding to the Gas5a GREM [110]. Indeed, the ERα E203G mutant binds GREs and Gas5a GREM, whereas the corresponding GRα G349E mutation reduces binding to both [110]. Another factor that contributes to the inability of the ERα DBD to bind Gas5 is that basic residues in H4 are not well conserved outside of the 3-ketosteroid receptors (GR, MR, PR, AR and ER) [112]. This explains why the transcriptional activity of not only ERα but also of another NR family member, PPARδ, when fused with the GAL4 DBD, was not affected by Gas5a lncRNA in vitro [109].

Although the GR-DBD binds to DNA and RNA with similar affinity, the mode of binding is distinct between the two. A molecular dynamics study based on DNA-free GR DBD (PDB ID: 6CFN), revealed that all GR DBD residues involved in RNA-binding were also involved in DNA-binding [113]. Thus, GR DBD binding to DNA and RNA involves similar residues but distinct protein–nucleic acid contacts [112,113]. The DNA-reading helix (H1) was more involved in DNA binding but also bound RNA [112]. Alanine substitutions of positively charged arginine and lysine residues in the H1 mutant (K442A/K446A) resulted in a 20-fold decrease in affinity for DNA but only a 4-fold for RNA [112]. The distorted helix (H2, see Figure 5) was involved in both DNA and RNA binding, and the corresponding mutant K467A/R470A displayed dramatically reduced binding for both [112]. Likewise, alanine substitutions of lysine and arginine in H3 had more pronounced effects on DNA binding [112]. Finally, H4 was reported as both folded [54,98] and unfolded [93,100] in crystal structures. Molecular dynamics simulations indicate that unfolding of H4 into a random-coil structure increases the number of electrostatic interactions with RNA [113].

### 3.3. Carboxy-Terminal Ligand-Binding Domain (LBD)

In 2002, Bledsoe et al. reported the first crystal structure of the F602S GR LBD in complex with dexamethasone and a peptide from the transcriptional intermediary factor 2 (TIF2) coactivator (see an overview of GR DBD crystal structures in Table 2) [142]. This structure revealed 11 α-helices and 4 β-strands that fold into three parallel layers to form an alpha helical sandwich, consistent with the LBD structure of other NRs (Figure 12) [63]. Helices 1 (H1) and H3 form the front, and H7, H10 and H11 form the back, whereas H4, H5, H8, and H9 form the middle layer of the domain [142]. This arrangement creates a hydrophobic cavity at the base of the receptor, termed the ligand-binding pocket (LBP) and is able to accommodate a variety of molecules [142]. The LBP volume ranges dramatically across the NR family from 30 Å^3^ in the *Drosophila* orphan nuclear receptor DHR38 to 1400 Å^3^ in the subtypes of peroxisome proliferator-activated receptors (PPARs) [143]. The top half of the LBD is conserved, while the bottom half is more variable, conferring ligand specificity among the NRs [144]. The volume of the GR LBP is approximately 600 Å^3^, while dexamethasone occupies only about 65% of the LBP [142]. To add to that, the pocket can expand to as much as 1070 Å^3^ as seen in the crystal structure of the deacylcortivazol-bound GR LBD [145]. Two distinct domains involved in coregulator binding are found in the LBD, the transcriptional activation function tau2 (τ2) region (residues 527 to 556) and the activation function 2 (AF2) region (Figure 3) [142,146].

#### 3.3.1. Agonist-Bound Form

GCs bound within the GR LBP form an extensive network of hydrogen bonds. The boundaries of this pocket are delineated by H3, H4/5, H7, H10/11 and H12 [142]. At one side of the pocket, the C_3_-ketone group of the GC A-ring forms a hydrogen bond with Arg611 of H5 and Gln570 of H3 (Figure 13) [142,147,158]. Another hydrogen bond forms between the C_11_ hydroxy group of the steroid C-ring and Asn564 of H3 [142,147,158]. The C_17_ hydroxy group of the D-ring interacts with Q642 of H7, while the C_20_ carbonyl and C_21_ hydroxy both form hydrogen bonds with T739 of H11 [142,147].

Although cortisol and dexamethasone are structurally similar, dexamethasone is much more potent and binds GR with much higher affinity [158]. The C_1_-C_2_ double bond observed in dexamethasone results in a planar steroid A-ring and C_3_ ketone group, facilitating the interaction of the latter with residues Arg611 and Gln570 [158]. In contrast, the C_1_-C_2_ single bond in cortisol is much more flexible which requires the steroid A-ring to bend in order to form a hydrogen bond with these residues [158]. Prednisolone, which is more potent than cortisol, exemplifies this as it is identical to cortisol except for the additional C_1_-C_2_ double bond [158]. It also appears that a water molecule is required for cortisol, but not dexamethasone, to form a hydrogen bond network to keep the ligand in place [158]. Interestingly, crystal structures of an ancestral variant of the GR (AncGR2) complexed with either cortisol or dexamethasone, revealed the presence of a water molecule for both ligands bound [166]. Dexamethasone also features two additional structural modifications, a fluorine atom at position C_9α_ and a methyl group at position C_16α_, which increase its interaction surface with the ligand-binding pocket [158].

In the dexamethasone-bound GR crystal structure (PDB ID: 1M2Z), a hydrophobic cavity, formed by helices 3, 5, 6, 7 and the β3-β4 turn above the steroid D-ring, is empty [158]. However, mometasone furoate (MF, PDB ID: 4P6W) has a C_17α_ furoate ester that occupies this hydrophobic cavity and makes hydrophobic interactions with Phe623, Ile629, Met639 and Cys643 which underlies this ligand’s higher affinity for GR [158]. The largest difference between the cortisol- or dexamethasone-bound vs. MF-bound GR LBP, is in Gln642 of H7 [158]. Gln642 is pushed away by the C_17α_ lipophilic group of MF, leading to an altered orientation of the C-terminus of the AF2-H, which is reported to be characteristic for high potency [158]. This increase in potency was also illustrated by a C_17α_ substitution to go from fluticasone propionate to the more potent fluticasone furoate (FF) [169].

Recently, the first crystal structure of a multidomain GR (DBD and LBD, residues 385–777) in complex with the agonists velsecorat or FF, a natural GRE (Sgk1) and a peptide of the PGC1α coactivator were resolved [122]. Both agonists interact with Asn564 to stabilize H12 in the active conformation (Figure 14). At the opposite side from Asn564, both ligands interact with Gln642, but the side chain is positioned differently due to the less bulky and non-steroidal scaffold of velsecorat. Most notable is the interaction of the 3-keto group of FF with Gln570 and Arg611, while velsecorat extends in a novel pocket beneath Trp577. Hydrogen–deuterium exchange mass spectrometry (HDX-MS) revealed that the two ligands led to different deuterium-uptake levels in a region of the DBD (residues 424–460, see Figure 5). This suggests there is ligand-specific communication between the LBD and DBD, even in the absence of DNA [122].

#### 3.3.2. Antagonist-Bound Form

Mifepristone, also known as RU-486, is a steroidal antiprogestin and antiglucocorticoid with weak antiandrogen activity [147,170]. The crystal structure of mifepristone in complex with a GR LBD (N517D, F602S, C638D) was first solved in 2003 (PDB ID: 1NHZ) [147]. In 2010, the crystal structure of the mifepristone-bound GR LBD in complex with a peptide from the corepressor NCOR1 (PDB ID: 3H52) was resolved [151]. Compared to other typical NR structures, the main differences were observed in H12 and in the binding of mifepristone and NCOR1 [151]. In the LBP, mifepristone adopts a conformation and orientation analogous to that of dexamethasone, albeit with a significant steric hindrance from the mifepristone moiety, which displaces H12, giving rise to three distinct monomers with distinct H12 conformations on different sides of the dimethylaminophenyl group (Figure 15) [151]. Hereafter, the three different H12 conformations in these monomers will be referred to as domains 1 to 3. The NCOR1 peptide is bound in domains 1 and 2, but not 3. In domain 1, H12 adopts a conformation that is remarkably different than the one observed in agonist-bound GR [151]. In domain 2, H12 adopts an intermediate conformation between that of domain 1 and the one GR acquires with an agonist [151]. Domain 1 is therefore likely to be the main antagonist conformation in which the adjusted position of H12 alters the coregulator binding site, thus increasing the affinity for the NCOR1 peptide [151]. Here, the charge clamp interaction of Lys579 from H3 with the C-terminus of NCOR1 is preserved, but the Glu755 interaction from H12 is lost [151].

An intriguing crystal structure is the one from *Heterocephalus glaber* (naked mole rat) GRβ LBD in complex with RU-486 (PDB ID: 5UC1) [163]. The general structure is very similar to that of GRα LBD/RU-486, and the in silico calculated total binding energies are similar, albeit some differences in binding interactions are observed [163]. In GRβ, the 50 C-terminal AAs of GRα are replaced by 15 unique residues resulting in the absence of H11 and H12 and appearance of a disordered C-terminal region (Figure 1) [46,163]. These unique C-terminal residues were found to be mostly disordered and not implicated in RU-486 binding. Remarkably, GRβ/RU-486 binds the NCOR corepressor with slightly higher affinity than GRα/RU-486 suggesting that H12 might be dispensable for corepressor binding [163]. This is further supported by the observation that H12 is positioned away from the AF2 region when the nonsteroidal antagonist compound 8 is bound (PDB ID: 4MDD) [157].

At the time of writing, no published studies are available for another structure involving RU-486 deposited in the Protein Data Bank with PDB ID 5UC3 [133].

A recent study by the team of Estebanez-Perpiña thoroughly reinvestigated GR LBD homodimerization, revealing distinct assemblies for antagonist-bound GR LBD compared to agonist-bound GR LBD [171]. In the antagonist-bound structures 5UC3, 1NHZ [147] and 3H52 [151] (Table 2), symmetric back-to-back and base-to-base homodimers were observed with much larger interface areas and lower energies compared to other GR LBD conformations. Antagonist binding is suggested to promote or stabilize these back-to-back and base-to-base assemblies, which is associated with an inactive or self-repressed state of GR. They also suggested that the deficiency in transcriptional activity of antagonist-bound GR may be attributed to impaired tetramerization, based on results from [172] in which RU-486-bound full-length GR is dimeric in the nucleus but unable to form tetramers on DNA [171].

The C-terminal H12 (also known as the activation function helix, AF2-H) forms the AF2 surface together with H3 and H4 and plays a major role in ligand-dependent interaction with coregulators [122]. The antagonist-bound GR has the AF2-H positioned in such a way that corepressor interactions are allowed [142]. Amongst the corepressors are NCOR1 and NCOR2 (or NCOR and SMRT, respectively) [173,174]. In agonist-bound GR, the AF2-H packs against helices 3, 4 and 10, stabilizing the receptor in the active conformation and promoting the association with coactivators proteins [142] such as NCOA1, 2, 3 and PPARγ coactivator 1α (PGC1α) [122,175,176]. Notably, some coregulators have the potential to act as either a coactivator or a corepressor, depending on the context, as was first shown for NCOA2 through the employment of distinct NCOA2 surfaces [177,178]. Binding of coregulators to the AF2 region occurs through a short amphipathic helix containing the LXXLL motif in coactivators or the (L/I)XX(I/V)I or LXXX(I/L)XXX(I/L) motif in corepressors [151,179,180]. A so-called charge clamp in which AA residues on the surface of GR interact with oppositely charged residues of the coregulator results in the formation of a stable complex [142]. In coactivator binding, a conserved charge clamp is mediated by K579 from H3 and E755 from H12, while a secondary charge clamp is mediated by R585 and D590 [142].

### 3.4. GR in Complex with Hsp

In the absence of GCs, apo-GR resides predominantly in the cytoplasm in association with a chaperone complex [181,182]. GR cycles through a four-step chaperoning process, which is essential for regulating its activity, stability, and ligand binding, and ensuring proper cellular response to GC hormones. In the first step, the active GR LBD is inactivated by Hsp70, upon which the cochaperone Hop (Hsp90-Hsp70 organizing protein) helps to load the Hsp70-GR complex onto Hsp90, forming an inactive ‘client-loading’ complex [168]. After hydrolysis of ATP by Hsp90 and closure of Hsp90, the ‘client-loading’ complex releases Hsp70 and Hop and incorporates p23 to form the ‘client-maturation’ complex, which restores GR ligand binding with increased affinity [167,168].

Two cryo-electron microscopy structures are reported: the ‘client-loading’ complex of Hsp90-Hsp70-Hop-GR [168] and the ‘client-maturation’ complex of Hsp90-p23-GR [167]. A model is proposed in which Hsp70C (Hsp70 client loading, the first Hsp70) captures the GR pre-H1 strand (residues 523–531 just N-terminal to/upstream of H1) causing the subsequent helix-strand motif to detach, thus destabilizing the GR LBP [168]. The partially unfolded GR is then delivered by Hsp70C to a complex containing Hop, Hsp70S (Hsp70 scaffolding, the second Hsp70) and an Hsp90 dimer producing the ‘client-loading’ complex. In this complex, GR is further unfolded by engagement of the GR H1 LXXLL motif (residues 532–536) with Hop and the GR post-H1 strand (residues just C-terminal/downstream of H1) with the Hsp90 lumen, impeding ligand binding [168]. Both Hsp70 proteins (Hsp70C and Hsp70S) and Hop are then released in a process requiring ATP hydrolysis and p23 is incorporated, forming the ‘client maturation’ complex in which H1 docks back onto the GR body and ligand binding is restored and enhanced [167,168]. This results in a mature apo-GR complex consisting of an Hsp90 dimer, p23 and one of the tetratricopeptide repeat (TRP)-containing cochaperones. Members belonging to the TRP-containing chaperones include FK506-binding protein (FKBP) 51 and 52 (two immunophilins), cyclophilin 40 and protein phosphatase (PP) 5. Ligand binding to GR is known to be inhibited by FKBP51, whereas FKBP52 is essential for cytoplasmic transport of liganded GR to the nucleus [183,184,185,186]. Upon hormone binding to GR, FKBP51 is replaced by FKBP52 [187]. Subsequent nuclear transport of GR along the microtubules is mediated by the interaction of FKBP52 and PP5 with the retrograde motor protein dynein [188]. At the nuclear membrane, nuclear import is mediated through interactions with components of the nuclear pore complex. Two nuclear localization signals (NLSs) have been identified in the GR sequence [189]. NLS1 overlaps with and extends C-terminally from the receptor DBD, and NLS2 is located within the LBD. The passage of GR through the nuclear pore complex starts with the recognition of NLS1 by the adaptor protein importin-α [190] and formation of a trimeric complex with importin-β [191]. Hsp90 has been shown to interact with importin-β and Nup62 [192]. Nuclear retention of GR is facilitated by a nuclear retention signal (NRS) which overlaps with NLS1 and delays nuclear export [193]. On the other hand, nuclear export of GR is mediated through the binding of its nuclear export signal (NES), located between the two zinc fingers in the DBD, to exportin-1 and calreticulin [194,195,196]. The cellular localization of the GR involves a dynamic process where both active and inactive forms have been shown to shuttle between the nucleus and cytoplasm [191,197]. Still, apo-GR is predominantly in the cytoplasm, whereas ligand-bound GR is predominantly in the nucleus.

## 4. GR Dimerization and Oligomerization

In 1983, the team of Yamamoto employed electron microscopy to study the behavior of the 94 kDa GR in the presence and absence of DNA [198]. They observed that the GR formed complexes of various sizes, ranging from homo-trimers to homo-hexamers, regardless of DNA binding [198]. Later studies proposed a dimeric mode of DNA binding and gene expression, guided by the first crystal structure of the GR DBD [93] and the identification of a 5 bp region (later termed the D-loop) in the DBD implicated in dimerization [199].

From a series of GR mutagenesis studies, it was posited that the GR-activating properties via direct dimeric binding were distinct and separable from GR’s gene-repressing properties, which are conferred by monomeric binding [127]. This was built on the observation that a D-loop mutant (A458T or GR_dim_) failed to activate genes while still being able to repress AP-1 [127]. Subsequent work involving this mutant in so-called GR^dim/dim^ mice recapitulated the hypothesis for endogenous GR target genes and demonstrated the mice’s viability, unlike the GR-null mice. This provided the basis to pursue the dissociated model of GR action and spurred the quest for specific so-called ‘dissociating’ ligands [126]. According to this model, ligands that could drive wild-type GR towards a GR_dim_ phenotype would attenuate inflammation, the main goal of GC therapy, via monomeric GR-repressing AP-1 and NF-κB while avoiding the GRE-driven clinically undesirable ‘side effects’ [200,201]. Such intensely sought after selective GR agonists (SEGRAs) and selective GR modulators (SEGRMs), collectively denominated SEGRAMs, are compounds designed to engage only a subset of GR-driven mechanisms in an attempt to reduce the number of side effects these compounds may have in the clinic [202]. SEGRA is the term that was first used for compounds derived from a steroidal scaffold, but unfortunately, these molecules had poor selectivity between steroid receptors [202]. The term SEGRM came later and was used to refer to the newer generation, including also non-steroidal compounds [202].

The simplified dissociated model of GR action has since been repeatedly challenged [203]. For instance, it was found that GR_dim_ was still able to dimerize and activate certain genes [54,69,95,129]. On the other hand, many anti-inflammatory actions of GCs were mediated by dimeric GR acting at conventional GREs, further disconnecting the binary vision of GR molecular biology from the complexity of ‘desirable’ vs. ‘adverse’ effects of GCs in vivo.

In the meantime, the structural basis of GR oligomerization continues to be debated. The first publication of the GR LBD crystal structure described an LBD dimer interface involving the loop between H1 and H3, the antiparallel β-sheet (formed by β3 and β4) and the C-terminal end of H5 [142]. A more recent in silico study analyzed the dimer interfaces of 21 GR LBD crystal structures deposited in the PDB and found that the interface observed by Bledsoe et al. was present in only six PDB entries [142,204]. The interface reported most often (in nine crystal structures) is the one involving H1, whereas the architecture with H9 in an anti-parallel arrangement is the most energetically favored, despite it only being observed once [204].

In the first multi-domain crystal structure of the GR LBD and DBD in complex with the agonist velsecorat, a GRE from *Sgk1* and a PGC1α coregulator peptide (residues 134–154), the dimerization interface reported by *Bledsoe* et al. was not observed and neither was the anti-parallel H9 interface described by *He* et al. and reported as most stable by Bianchetti et al. [122,142,158,204]. Instead, two possible LBD dimer interfaces were proposed [122]. The first interface is a head-to-tail dimer with a buried surface area of 904.4 Å^2^ involving the N-terminal end of H10 and H11 and H6-H7 and this interface partially overlaps with the canonical ER dimer interface [122]. The other is a head-to-head dimer primarily mediated by H1, whereby only 710.2 Å^2^ is buried and which is most often observed in crystal structures [122,204].

While the H9–H10–H11 dimerization interface of ERα and the estrogen-related receptors (ERRs) is well-established, this is not the case for the 3-ketosteroid receptors [204]. The C-terminal residues of GR form the F-domain, which packs against the LBD and prevents H9–H10–H11 dimerization, as observed in ERα and the ERRs [204]. Comparing the interfaces available in the PDB reveals that to date, there is no consensus on how the GR LBD dimerizes [133]. Moreover, the concept that GR either acts as a monomer or a dimer has been challenged when DNA binding was proposed to induce the formation of GR tetramers [171,172,205]. A recent study by Jiménez-Panizo et al. describes four GR agonist-bound modes of dimerization, centered around the residues Tyr545 and Ile628, and these would further generate different tetrameric arrangements on DNA [171]. An additional observation was that D641V, a GR mutant linked to Chrousos syndrome [206], formed higher-order oligomers on DNA, but this was accompanied by reduced transcriptional activity [171]. The authors suggest that the D641V mutation promotes the formation of a non-functional multimer, explaining the GC-resistant phenotype [171].

In conclusion, the study of the GR is far from over. Despite significant progress in understanding the structural and functional aspects of the receptor, there are still many debates and controversies to be settled.

## 5. Conclusions and Future Perspectives

The advancements made in understanding the multidomain structure of the GR have provided a critical framework for understanding its signaling and how this drives the activation and repression of genes in response to different ligands, through the binding of distinct DNA sequences and interaction with various coregulators. However, our insights also come with limitations. X-ray diffraction studies often involve stabilizing mutations, and although these are chosen in such a way that interference with the wild-type structure is minimal, one cannot be certain of such a claim. Additionally, crystal structures often involve individual domains of the GR, and therefore, information on domain interplay is lacking. Recently, however, the first GR multidomain (DBD and LBD) crystal structure was resolved and hopefully more will follow soon, allowing us to improve our understanding of GR domain interplay and its behavior on distinct DNA sequences, with different coregulators in response to different ligands. Much can also be learned from studying other NRs or by solving GR heterodimers, for example, the GR-MR heterodimer, which is recognized as physiologically relevant [207,208,209,210]. Ultimately, these insights will aid in the search for improved GC therapies 

## Figures and Tables

**Figure 1 cells-12-01636-f001:**
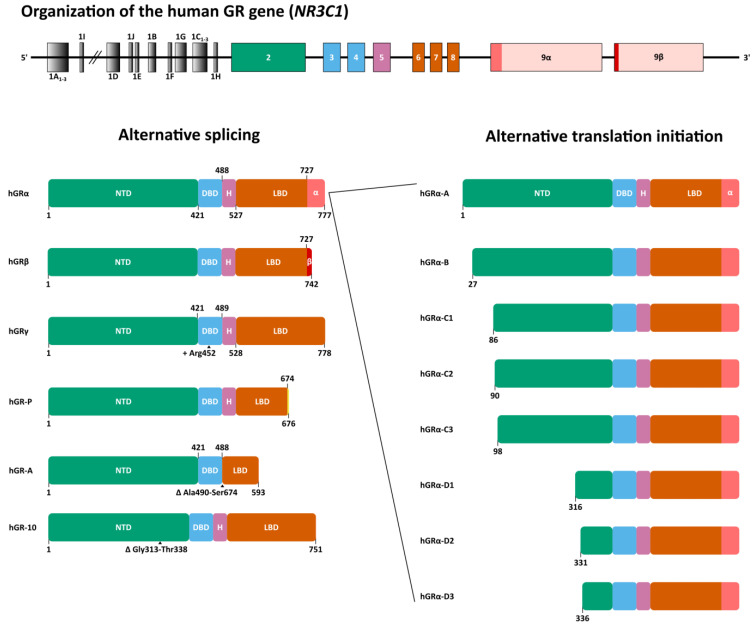
Organization of the human glucocorticoid receptor (hGR) gene (*NR3C1*) and alternative splice and translation initiation variants of the hGR protein. GR, glucocorticoid receptor; NTD, N-terminal domain; DBD, DNA-binding domain; H, Hinge region; LBD, Ligand-binding domain.

**Figure 2 cells-12-01636-f002:**
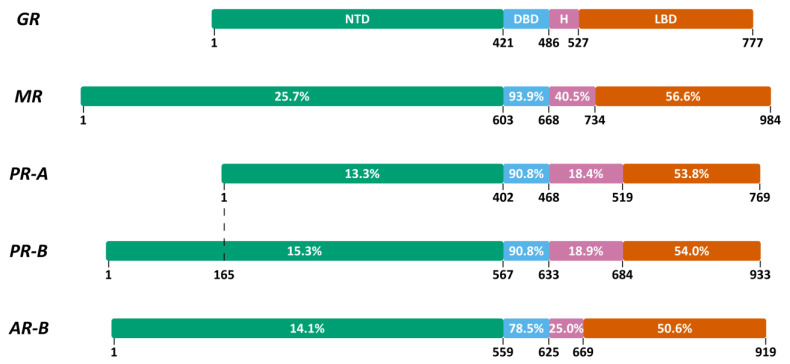
3-ketosteroid receptors and their homologies expressed as percent identity to hGR. GR, glucocorticoid receptor; MR, mineralocorticoid receptor; PR-A, progesterone receptor A; PR-B, progesterone receptor B; AR-B, androgen receptor B; NTD, N-terminal domain; DBD, DNA-binding domain; H, hinge region; LBD, ligand-binding domain.

**Figure 3 cells-12-01636-f003:**
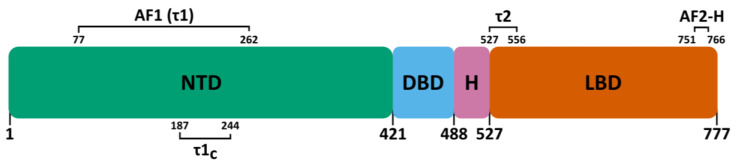
Linear domain structure of the hGR protein. Domains from left to right: N-terminal domain (NTD), DNA-binding domain (DBD), hinge region (H) and ligand-binding domain (LBD). Three additional regions are shown: activation function 1 (AF1), tau1 core (τ1_C_), tau2 (τ2), activation function 2 helix (AF2-H or helix 12).

**Figure 4 cells-12-01636-f004:**
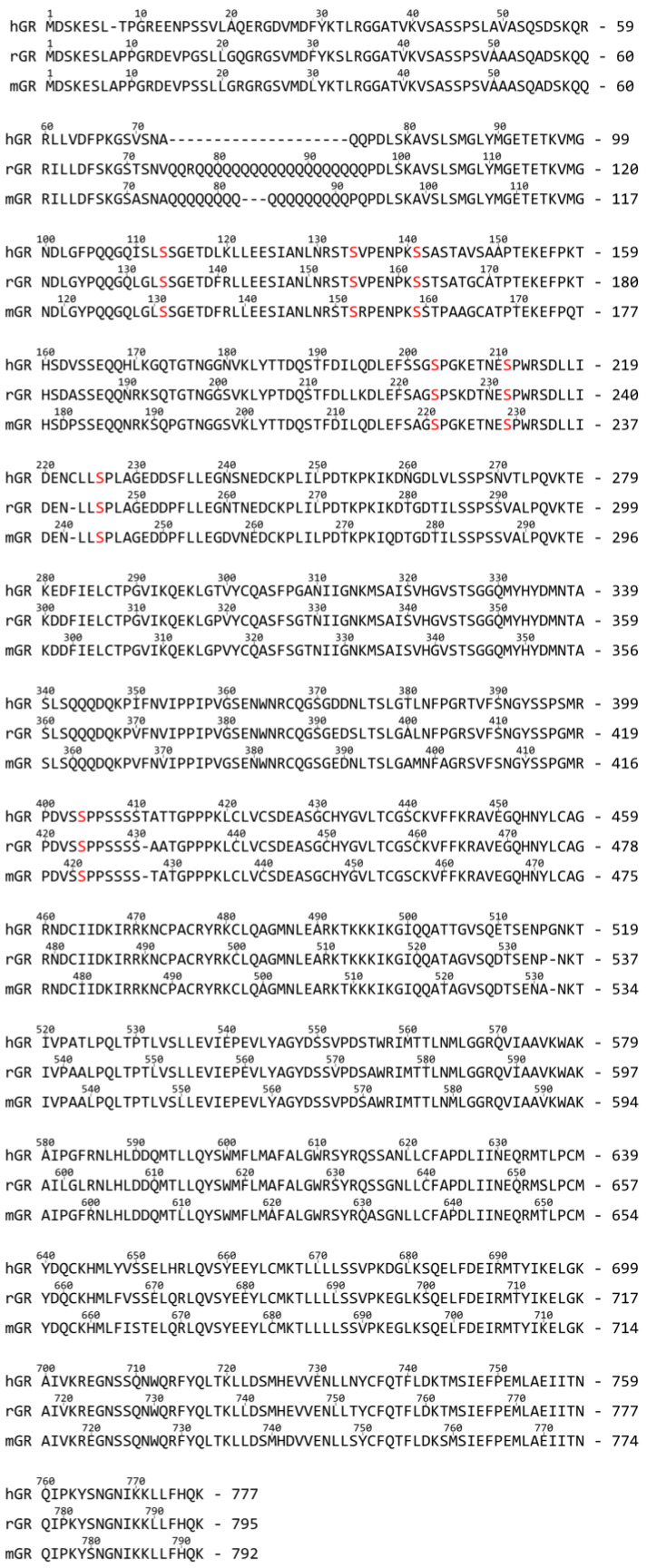
Cross-species protein sequence alignment of GR. Comparison of GR from *Homo sapiens* (hGR; UniProt ID P04150), *Rattus norvegicus* (rGR; UniProt ID P06536) and *Mus musculus* (mGR; Uniprot ID P06537). The letters S in red mark the Serines listed in referral to this Figure.

**Figure 5 cells-12-01636-f005:**
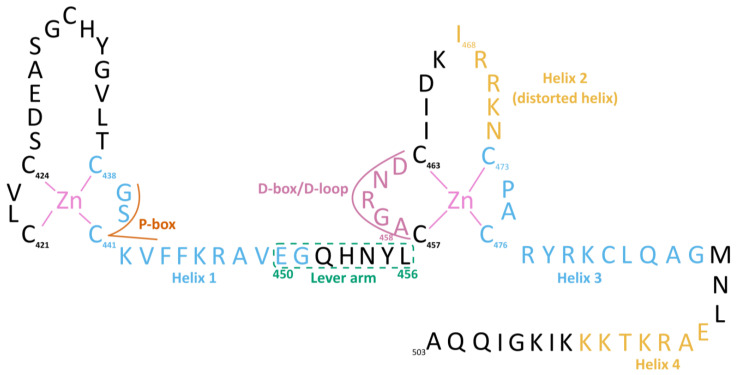
Sequence of the hGR DBD and part of the hinge region (residues 421–503). The DBD contains two zinc fingers, the proximal box (P-box), helix 1 (H1), the lever arm, the D-box or D-loop, a distorted helix 2 (H2), helix 3 (H3) and helix 4 (H4). Helix positions are based on the PDB ID 3G99 crystal structure and are estimates as these are of dynamic nature [54].

**Figure 6 cells-12-01636-f006:**
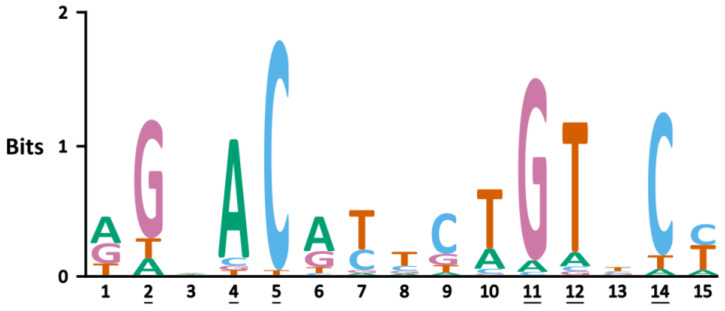
Sequence logo for the observed GR binding motif by Watson et al. Underlined positions indicate positions of bases directly contacted by GR. Figure adapted from [95].

**Figure 7 cells-12-01636-f007:**
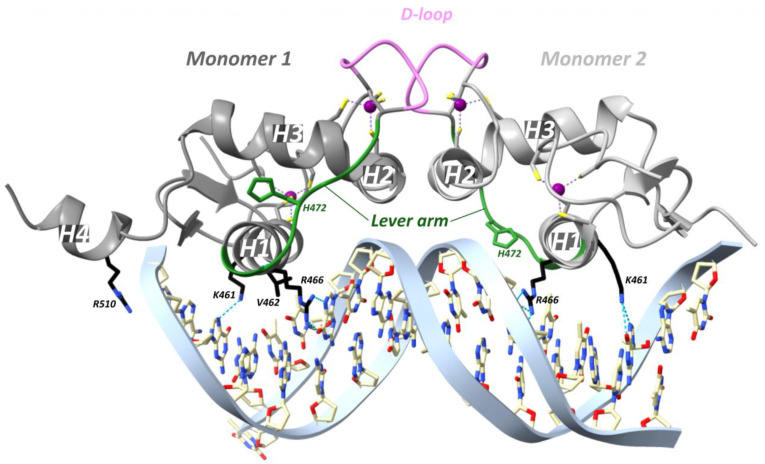
Crystal structure of the rat GR DBD and the Pal complex-9 GRE (PDB ID: 3G99) [54]. Residues Lys461, Val462 and Arg466 from the first GR monomer and residues Lys461 and Arg466 from the second monomer make specific DNA major groove contacts. Arg510 mediates a non-specific contact in the DNA minor groove. In the first monomer, His472 is seen packed in the core of the protein fold, while in the second monomer, it is flipped out. Hydrogen bonds are shown as blue dashed lines; zinc ions are depicted as purple spheres. H1, helix 1; H2, helix 2; H3, helix 3; H4, helix 4.

**Figure 8 cells-12-01636-f008:**
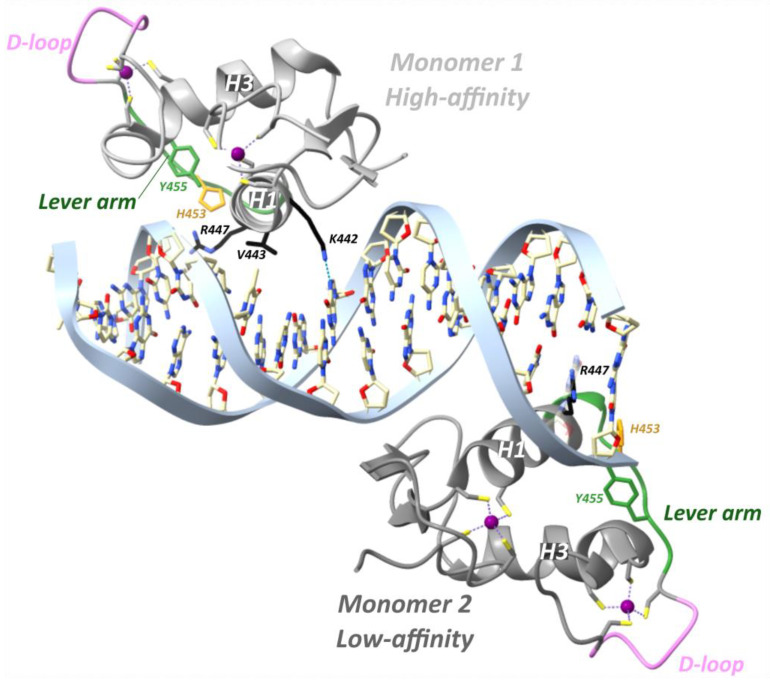
Crystal structure of the hGR DBD and the TSLP IR-nGRE (PDB ID: 4HN5) [100]. Residues Lys442, Val443 and Arg447 from the first GR monomer and residue Arg447 from the second monomer make specific DNA major groove contacts. His453 is flipped out from the core of the protein fold in both monomers and stabilized by Tyr455 and Arg447. Hydrogen bonds are shown as blue dashed lines; zinc ions are depicted as purple spheres. H1, helix 1; H3, helix 3.

**Figure 9 cells-12-01636-f009:**
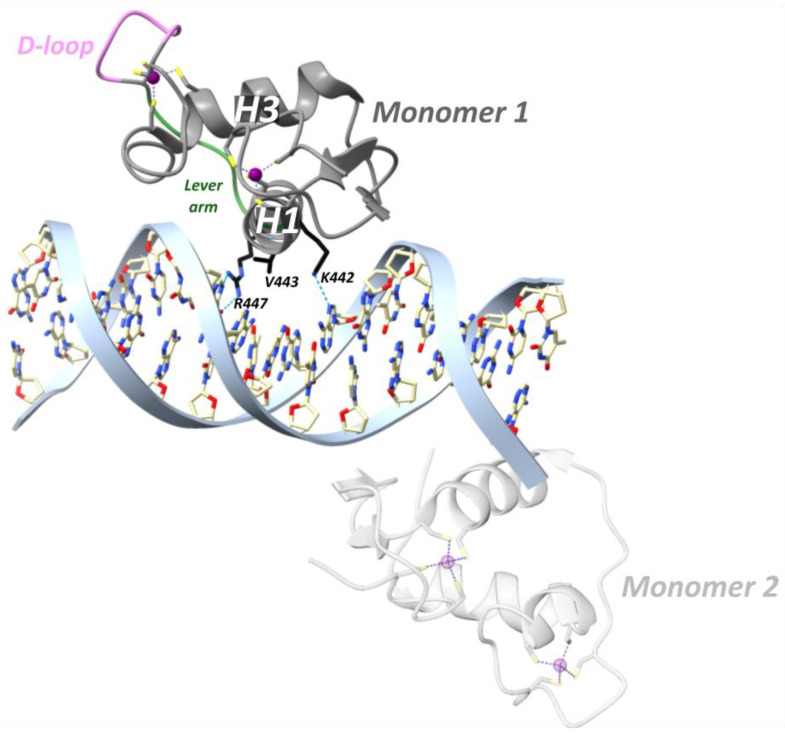
Crystal structure of the human GR DBD and the IL-11 TRE (PDB ID: 5VA7) [104]. Hydrogen bonds are shown as blue dashed lines; zinc ions are depicted as purple spheres. H1, helix 1; H3, helix 3.

**Figure 10 cells-12-01636-f010:**
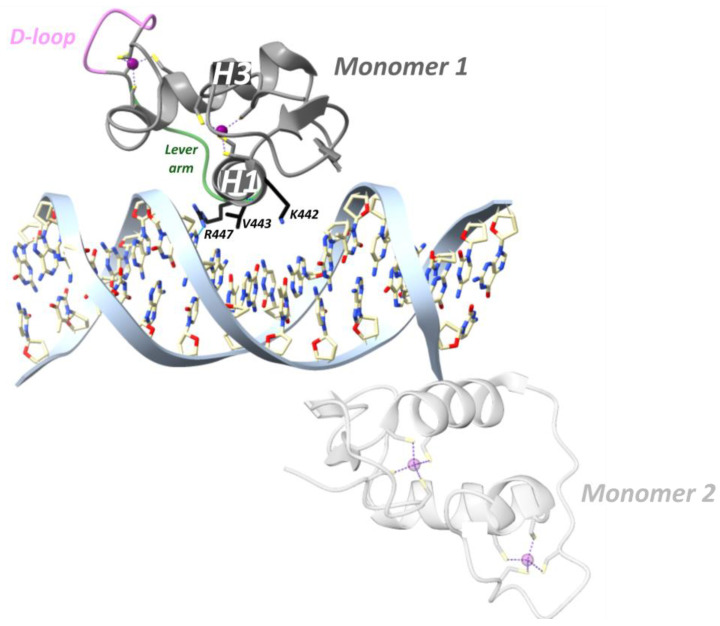
Crystal structure of the human GR DBD and the VCAM-1 TRE (PDB ID: 5VA0) [104]. Hydrogen bonds are shown as blue dashed lines; zinc ions are depicted as purple spheres. H1, helix 1; H3, helix 3.

**Figure 11 cells-12-01636-f011:**
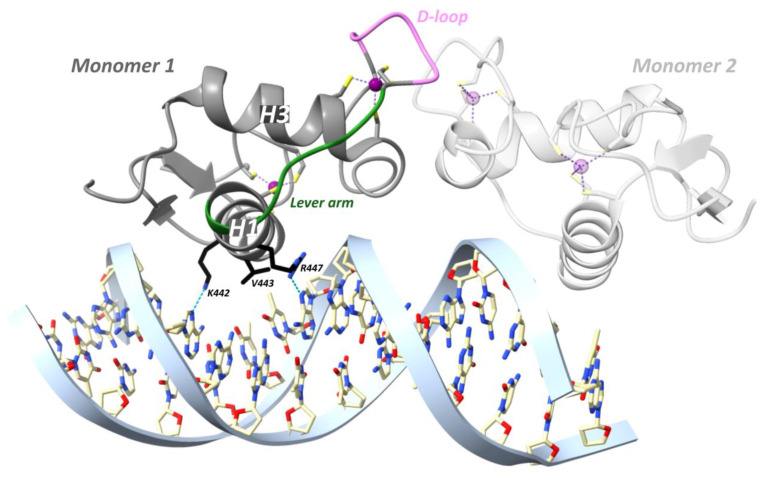
Crystal structure of the human GR DBD and the IL-8 κBRE (PDB ID: 5E69) [102]. Hydrogen bonds are shown as blue dashed lines; zinc ions are depicted as purple spheres. H1, helix 1; H3, helix 3.

**Figure 12 cells-12-01636-f012:**
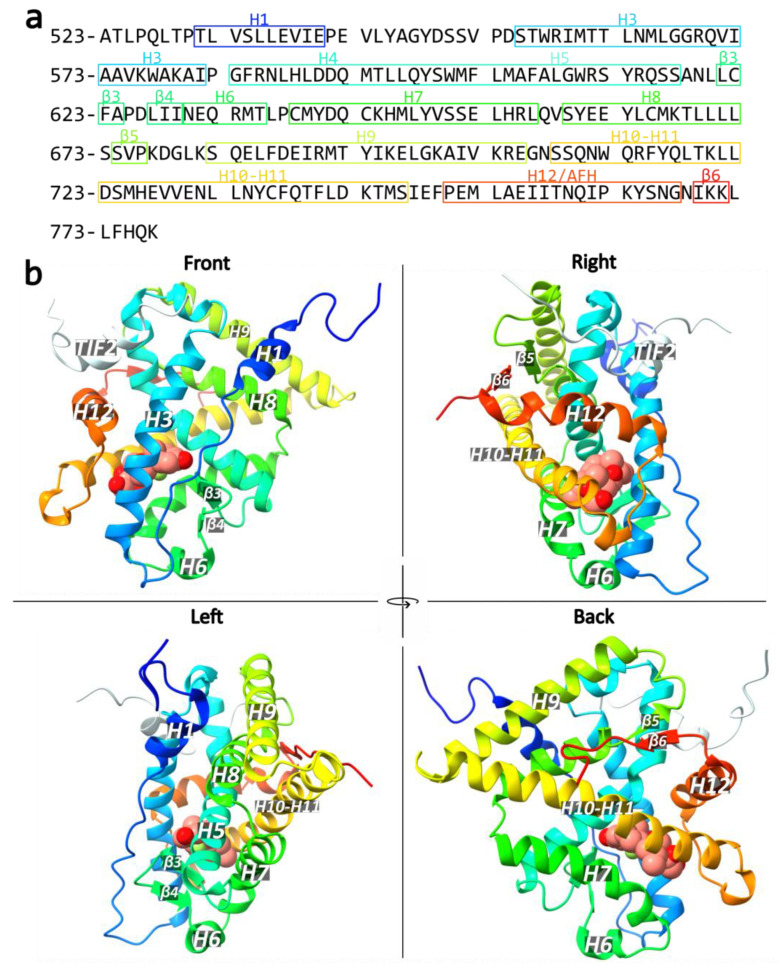
(**a**) Protein sequence of the GR LBD. (**b**) Structural overview of the GR LBD in complex with dexamethasone and a TIF2 coactivator motif (PDB ID: 1M22) [142]. Dexamethasone is depicted as a sphere colored in red/salmon.

**Figure 13 cells-12-01636-f013:**
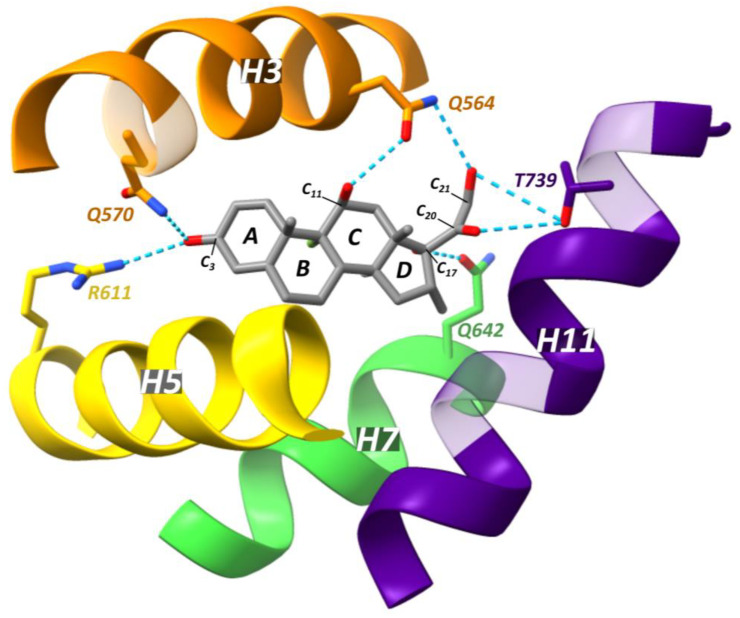
Structural overview of dexamethasone (gray) bound in the GR LBP with hydrogen bonds shown as blue dashed lines (PDB ID: 1M2Z) [142]. The C_3_ ketone from the A-ring interacts with Gln570 and Arg611 belonging to H3 (orange) and H5 (yellow), respectively. The C_11_ hydroxy group from the C-ring interacts with Asn564 belonging to H3 (orange). The C_17_ hydroxy group from the D-ring interacts with Q642 belonging to H7 (green). The C_20_ carbonyl and C_21_ hydroxy group interact with Thr739 belonging to H11 (purple).

**Figure 14 cells-12-01636-f014:**
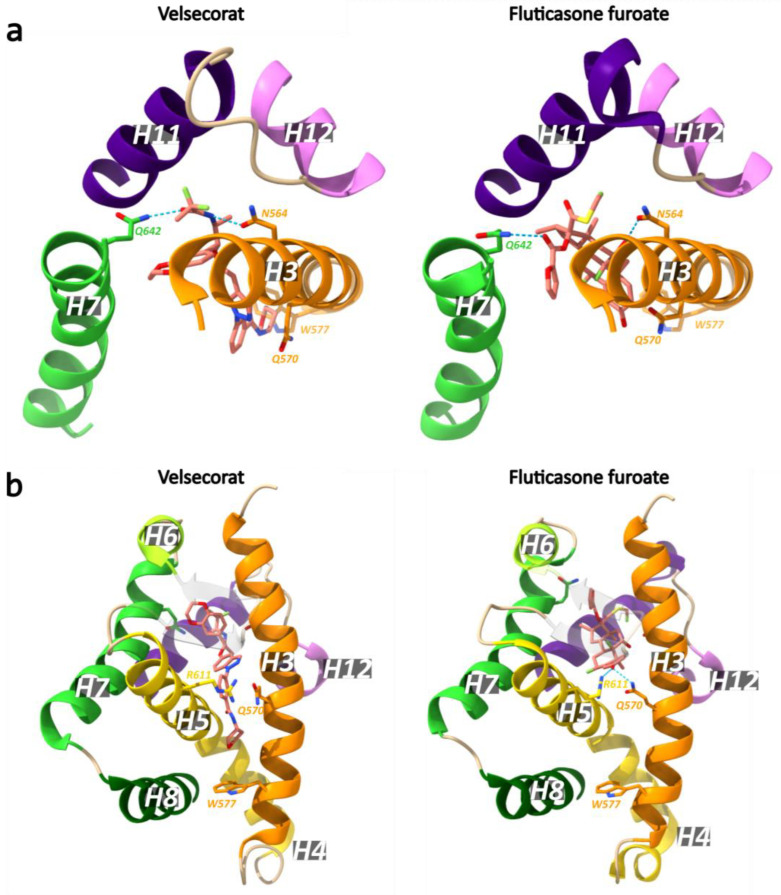
Structural overview of velsecorat and fluticasone furoate (FF) bound in the GR LBP with hydrogen bonds shown as blue dashed lines (PDB ID: 7PRW and 7PRV, respectively) [122]. (**a**) Both ligands interact with Asn564 and Gln642 to stabilize H12. (**b**) Velsecorat extends in a novel pocket beneath Trp577, while FF interacts with Arg611 and Gln570.

**Figure 15 cells-12-01636-f015:**
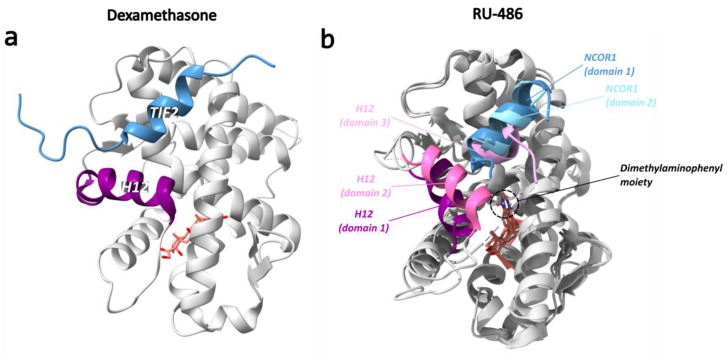
Structural overview of the dexamethasone-bound and RU-486-bound GR LBD (PDB ID: 1M2Z and 3H52, respectively) [142,151]. (**a**) Agonist-bound GR LBD as reference with H12 (purple), TIF2 (blue) and dexamethasone (salmon) colored. (**b**) Overlay of the three domains (domain 1–3) of RU-486-bound GR LBD. In domain 1, RU-486 (salmon) displaces H12 (purple) from the agonist position, enabling the binding of NCOR1 (dark blue). In domain 2, H12 (pink) is observed in an intermediate position between the positions observed in domain 1 and the agonist-bound GR LBD. In domain 3, H12 (light pink) is observed on the other side of the dimethylaminophenyl moiety, occupying the coregulator binding site and thus preventing NCOR1 binding.

**Table 1 cells-12-01636-t001:** Overview of available crystal structures of the GR DBD, retrieved from the Protein Data Bank (PDB).

PDB ID	GR Region + Mutations	Species	GRE Sequence	Method	Ref.
1GDC	439–510 /	*R. norvegicus*	/	Solution NMR	[96]
1GLU	440–525 /	*R. norvegicus*	CCAGAACATCGATGTTCTG (Consensus GRE with 4 nt spacer)	XRD	[93]
1LAT	440–515 G458E, S459G, V462A, A477K, G478Y, R479E, N480G, D481K	*R. norvegicus*	TTCCAGAACATGTTCTGGA	XRD	[99]
1R4O	440–525 /	*R. norvegicus*	CCAGAACATCGATGTTCTG (Consensus GRE with 4 nt spacer)	XRD	[93]
1R4R	440–525 /	*R. norvegicus*	TCAGAACATGATGTTCTCA	XRD	[93]
1RGD	440–510 /	*R. norvegicus*	/	Solution NMR	[97]
2GDA	/	*R. norvegicus*	/	Solution NMR	[96]
3FYL	440–525 /	*R. norvegicus*	AAGAACATTTTGTCCG	XRD	[54]
3G6P	440–525 /	*R. norvegicus*	CCAGAACACCCTGTTCTG (FKBP5 18 bp)	XRD	[54]
3G6Q	440–525 /	*R. norvegicus*	TAGAACAGGGTGTTCT (FKBP5 binding site complex 9)	XRD	[54]
3G6R	440–525 /	*R. norvegicus*	CCAGAACAGGGTGTTCTG (FKBP5 complex-52 18 bp)	XRD	[54]
3G6T	440–525 G470^Q471insR (GRγ)	*R. norvegicus*	AAGAACAGGGTGTTCT (FKBP5 16 bp complex-34)	XRD	[54]
3G6U	440–525 /	*R. norvegicus*	AAGAACACCCTGTTCT (FKBP5 16 bp complex-49)	XRD	[54]
3G8U	440–525	*R. norvegicus*	AAGAACATTGGGTTCC (GILZ 16 bp complex-5)	XRD	[54]
3G8X	440–525	*R. norvegicus*	AAGAACATTGGGTTCC (GILZ 16 bp complex-65)	XRD	[54]
3G97	440–525	*R. norvegicus*	TGGAACCCAATGTTCT (GILZ 16 bp complex-9)	XRD	[54]
3G99	440–525	*R. norvegicus*	AAGAACATTTTGTTCT (Pal complex-9)	XRD	[54]
3G9I	440–525	*R. norvegicus*	AAGAACATTTTGTTCT (Pal complex-35)	XRD	[54]
3G9J	440–525	*R. norvegicus*	CCAGAACAAAATGTTCTG (Pal, 18 bp complex-36)	XRD	[54]
3G9M	440–525	*R. norvegicus*	AAGAACATTTTGTCCG (Sgk, 16 bp complex-44)	XRD	[54]
3G9O	440–525	*R. norvegicus*	AAGAACATTTTGTCCG (Sgk, 16 bp complex-9)	XRD	[54]
3G9P	440–525 /	*R. norvegicus*	AAGAACATTTTGTCCG (Sgk 16 bp complex 7)	XRD	[54]
4HN5	417–506	*Homo sapiens*	CGCCTCCGGGAGAGCT (TSLP IR-nGRE)	XRD	[100]
4HN6	417–506 R460D, D462R	*H. sapiens*	CGCCTCCGGGAGAGCT (TSLP IR-nGRE)	XRD	[100]
5CBX	412–495	Ancestral	CCAGAACAGAGTGTTCTG	XRD	[101]
5CBY	412–495	Ancestral	CCAGAACAGAGTGTTCTG	XRD	[101]
5CC1	412–495 S425G	Ancestral	CCAGAACAGAGTGTTCTG	XRD	[101]
5E69	417–506	*H. sapiens*	ATCGTGGAATTTCCTC (IL-8 ĸB-RE)	XRD	[102]
5E6A	417–506	*H. sapiens*	ATCAGGAAATTCCCAG (PLAU ĸB-RE)	XRD	[102]
5E6B	417–506	*H. sapiens*	CCGGGGAATTCCGCCG (RelB ĸB-RE)	XRD	[102]
5E6C	417–506	*H. sapiens*	AGTGGAAATTCCCACT (CCL2 ĸB-RE)	XRD	[102]
5E6D	417–506	*H. sapiens*	GCTCCGGAATTTCCAA (ICAM-1 ĸB-RE)	XRD	[102]
5EMC	411–500	*H. sapiens*	CCAGAA(methyl)CATCATGTTCTG	XRD	[103]
5EMP	411–500	*H. sapiens*	CCAGAACATGATGTTCTG	XRD	[103]
5EMQ	411–500	*H. sapiens*	CCAGAACATCATGTTCTG	XRD	[103]
5VA0	419–490	*H. sapiens*	CGGCTGACTCATCAAG (VCAM-1 TRE)	XRD	[104]
5VA7	419–488	*H. sapiens*	AGGGTGAGTCAGGATG (IL-11 TRE)	XRD	[104]
6BQU	421–490	*H. sapiens*	AAGCTAGTACATTTGC (monomeric DNA binding site)	XRD	/
6BSE	420–505	*S. oedipus*	ACCACGTGTACTTTTT	XRD	/
6BSF	418–507	*H. sapiens*	AAGCTAGTACATTTGC	XRD	/
6CFN	418–506	*H. sapiens*	/	XRD	[98]
6 × 6D	417–490	*H. sapiens*	CCAGAACGGAGCGTTCTG (pre-GRE)	XRD	[105]
6X6E	417–491	*H. sapiens*	CCAGAACGGAG(methyl)CGTTCTG (methylated pre-GRE)	XRD	[105]

XRD, X-ray diffraction; NMR, nuclear magnetic resonance.

**Table 2 cells-12-01636-t002:** Overview of available crystal structures of the ligand-bound GR LBD, retrieved from the Protein Data Bank (PDB).

PDB ID	Ligand	GR Region + Mutations	Species	Cofactor Peptide	GRE Sequence	Method	Ref.
1M2Z	Dex	521–777 F602S	*H. sapiens*	PVSPKKKENALLRYLLDKDDT (NCOA2)	/	XRD	[142]
1NHZ	RU-486	500–777 N517D, F602S, C638D	*H. sapiens*	/	/	XRD	[147]
1P93	Dex	500–777 N517D, F602S, C638D	*H. sapiens*	KENALLRYLLDK (NCOA2)	/	XRD	[147]
3BQD	Deacyl-cortivazol (DAC)	525–777 F602S	*H. sapiens*	AQQKSLLQQLLTE (NCOA1)	/	XRD	[145]
3CLD	Fluticasone furoate (GW6)	521–777 F602Y, C638G	*H. sapiens*	KENALLRYLLDK (NCOA2)	/	XRD	[148]
3E7C	GSK866	521–777 F602Y, C638G	*H. sapiens*	ENALLRYLLDK (NCOA2)	/	XRD	[149]
3GN8	Dex	529–777 N.A.	*Ancestral*	PVSPKKKENARYLLDKDDT (NCOA2)	/	XRD	[150]
3H52	RU-486	528–777 F602S, C638D, E684A, E688A, W712S	*H. sapiens*	ASNLGLEDIIRKALMGSFD (NCOR1)	/	XRD	[151]
3K22	alaninamide 10	521–777 F602Y, C638G	*H. sapiens*	KENALLRYLLDK (NCOA2)	/	XRD	[152]
3K23	D-prolinamide 11	521–777 F602Y, C638G	*H. sapiens*	KENALLRYLLDK (NCOA2)	/	XRD	[152]
3MNE	Dex	527–783 F617S	*M. musculus*	KENALLRYLLDKD (NCOA2)	/	XRD	[153]
3MNO	Dex	527–783 F608S, A611V	*M. musculus*	KENALLRYLLDKD (NCOA2)	/	XRD	[153]
3MNP	Dex	527–783 A611V, V708A, E711G	*M. musculus*	KENALLRYLLDKD (NCOA2)	/	XRD	[153]
4CSJ	Compound 30 (NN7)	500–777 N517D, V571M, F602S, C638D	*H. sapiens*	ENALLRYLLDKDD (NCOA2)	/	XRD	[154]
4E2J	mometasone furoate	Synthetic 250AA fragment, ancestral GR	*Ancestral*	NCOA2 (741–752)	/		[155]
4LSJ	Compound 10 (LSJ)	522–777 F602Y, C638G	*H. sapiens*	HSSRLWELLMEAT (Synthetic D30 peptide)	/	XRD	[156]
4MDD	Compound 8 Non-steroidal antagonist	522–777 L525S, L528S, L535A, V538T, F602Y, C638D, E684A, E688A, W712S	*H. sapiens*	NLGLEDIIRKALMGS (NCOR1)	/	XRD	[157]
4P6W	Mometasone Furoate	526–777 F602A, C622Y, T668V, S674T, V675I, K699A, K703A	*H. sapiens*	ANALLRYLLDKD (NCOA2)	/	XRD	[158]
4P6X	Cortisol	523–777 F602A, C622Y, T668V, S674T, V675I, E684A, E688A	*H. sapiens*	KENALLRYLLDKDD (NCOA2)	/	XRD	[158]
4UDC	Dex	500–777 N517D, F602S, C638D	*H. sapiens*	KENALLRYLLDKDD (NCOA2)	/	XRD	[159]
4UDD	Desisobutyryl-ciclesonide	500–777 N517D, V571M, F602S, C638D	*H. sapiens*	KENALLRYLLDKDD (NCOA2)	/	XRD	[159]
5G3J	Compound 15 (E7T)	500–777 N517D, V571M, F602S, C638D	*H. sapiens*	KENALLRYLLDKDD (NCOA2)	/	XRD	[160]
5G5W	Compound 8b (R8C)	500–777 N517D, V571M, F602S, C638D	*H. sapiens*	KENALLRYLLDKDD (NCOA2)	/	XRD	[161]
5NFP	Budesonide	500–777 N517D, V571M, F602S, C638D,	*H. sapiens*	KENALLRYLLDKDD (NCOA2)	/	XRD	[162]
5NFT	AZD5423	500–777 N517D, V571M, F602S, C638D, E638A, W712S	*H. sapiens*	KENALLRYLLDKDD (NCOA2)	/	XRD	[162]
5UC1	RU-486	519–727 (GRβ) F599S	*H. glaber*	/	/	XRD	[163]
5UC3	RU-486	522–777 L733K, N734P	*H. sapiens*	/	/	XRD	/
6DXK	Compound 11	522–777 L525S, L528S, L535A, V538T, F602Y, C638D, E684A, E688A? W712S	*H. sapiens*	/	/	XRD	[164]
6EL6	Compound 4	500–777 N517D, V571M, F602S, C638D	*H. sapiens*	KENALLRYLLDKDD (NCOA2)	/	XRD	[165]
6EL7	Compound 31	500–777 N517D, V571M, F602S, C638D, E684A, W712S	*H. sapiens*	KENALLRYLLDKDD (NCOA2)	/	XRD	[165]
6EL9	AZD9567	500–777 N517D, V571M, F602S, C638D, E684A, W712S	*H. sapiens*	KENALLRYLLDKDD (NCOA2)	/	XRD	[165]
6NWK	Dex	529–777 N.A.	*Ancestral*	PSLLKKLLLAPA (PGC1α)	/	XRD	[166]
6NWL	Cortisol	529–777 N.A.	*Ancestral*	PSLLKKLLLAPA (PGC1α)	/	XRD	[166]
7KRJ	Dex	520–777 F602S	*H. sapiens*	Hsp90, p23 (full-length)	/	CEM	[167]
7KW7	/	Full-length (LBD structure)	*H. sapiens*	Hsp90, Hsp70, Hop (full-length)	/	CEM	[168]
7PRV	Fluticasone furoate (GW6)	385–777 S404A N517D V571M F602S C638D	*H. sapiens*	PPQEAEEPSLLKKLLLAPANT (PGC1α)	TACAGAACATTTTGTCCGTCGAC (Sgk1 23 bp; overhang)	XRD	[122]
7PRW	Velsecorat	385–777 S404A N517D V571M F602S C638D	*H. sapiens*	PPQEAEEPSLLKKLLLAPANT (PGC1α)	GTACAGAACATTTTGTCCGTCGA (Sgk1 23 bp; blunt)	XRD	[122]
7PRX	Velsecorat	529–777	*H. sapiens*	PPQEAEEPSLLKKLLLAPANT (PGC1α)	/	XRD	[122]

Dex, dexamethasone; NCOA2, nuclear receptor coactivator 2; XRD, X-ray diffraction; NCOA1, nuclear receptor coactivator 1; NCOR1, nuclear receptor corepressor 1; PGC1α, Peroxisome proliferator-activated receptor gamma coactivator 1 alpha; Hsp90, heat shock protein; CEM, cryo-electron microscopy; LBD, ligand-binding domain; Hsp70, heat shock protein 70, Hop, Hsp90-Hsp70 organizing protein.

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
