# Peer review of "The Biologist’s Guide to the Glucocorticoid Receptor’s Structure"

_cells, 2023, doi:10.3390/cells12121636_

Round 1

Reviewer 1 Report

In this manuscript, Deploey et al reviewed the multidomain structure of the GR alpha and discuss how this drives its biological response through binding to DNA sequences. From this perspective, the article is informative, well written and complete. Nonetheless, there is a missing point related to the mechanism of activation of the receptor by steroid binding followed by its retrotransport mechanism by the HSP90-FKBP52 chaperone system, as well as how the same molecular machinery regulates the nuclear retention of GR, which are essential features of its biological response. The roles of HSP90-binding immunophilins are not addressed. It is my humble suggestion to include these topics to make the article more complete for readers looking for the full picture of the mechanism of action of this receptor.

Author Response

Response to Reviewer 1 Comments

Point 1: In this manuscript, Deploey et al reviewed the multidomain structure of the GR alpha and discuss how this drives its biological response through binding to DNA sequences. From this perspective, the article is informative, well written and complete. Nonetheless, there is a missing point related to the mechanism of activation of the receptor by steroid binding followed by its retrotransport mechanism by the HSP90-FKBP52 chaperone system, as well as how the same molecular machinery regulates the nuclear retention of GR, which are essential features of its biological response. The roles of HSP90-binding immunophilins are not addressed. It is my humble suggestion to include these topics to make the article more complete for readers looking for the full picture of the mechanism of action of this receptor.

Response:

We cordially thank the reviewer for taking the time to review and for having pointed out a missing section, which we have now included to improve our manuscript. We have added the yellow marked section to the text below (new lines 808-829 in the manuscript, section 3.2 “GR in complex with Hsp”) and included extra references.

This results in a mature apo-GR complex consisting of an Hsp90 dimer, p23 and one of the tetratricopeptide repeat (TRP)-containing cochaperones. Members belonging to the TRP-containing chaperones include FK506-binding protein (FKBP) 51 and 52 (two immunophilins), cyclophilin 40 and protein phosphatase (PP) 5. Ligand binding to GR is known to be inhibited by FKBP51, whereas FKBP52 is essential for cytoplasmic transport of liganded GR to the nucleus [184–187]. Upon hormone binding to GR, FKBP51 is replaced by FKBP52 [188]. Subsequent nuclear transport of GR along the microtubules is mediated by the interaction of FKBP52 and PP5 with the retrograde motor protein dynein [189]. At the nuclear membrane, nuclear import is mediated through interactions with components of the nuclear pore complex. Two nuclear localization signals (NLSs) have been identified in the GR sequence [190]. NLS1 overlaps with and extends C-terminally from the receptor DBD and NLS2 is located within the LBD. The passage of GR through the nuclear pore complex starts with the recognition of NLS1 by the adaptor protein importin-α [191] and formation of a trimeric complex with importin-β [192]. Hsp90 has been shown to interact with importin-β and Nup62 [193]. Nuclear retention of GR is facilitated by a nuclear retention signal (NRS) which overlaps with NLS1 and delays nuclear export [194]. On the other hand, nuclear export of GR is mediated through binding of its nuclear export signal (NES), located between the two zinc fingers in the DBD, to exportin-1 and calreticulin [195–197]. The cellular localization of the GR involves a dynamic process where both active and inactive forms have been shown to shuttle between the nucleus and cytoplasm [198,192]. Still, apo-GR is predominantly in the cytoplasm, whereas ligand-bound GR is predominantly in the nucleus.

Reviewer 2 Report

The review article “The Biologist’s Guide to the Glucocorticoid Receptor’s Structure” by Deploey et. al. is well written and describe the advances in structural understanding of glucocorticoid receptor. I have few minor concerns and suggestion regarding this article. 

1. Line 207-208 – there is no reference for phosphorylation at Ser211.

2. Line 270-272 – coordination bond is not van der walls interaction.  

3. Table 1 – pdb id 1RGD – method is described as “solution NMR” which is fine but then in same table, there are other structures where the method is written as “NMR” giving the impression that the other structures are solved by different method like solid state NMR which is not correct. The authors need to be consistent.

4. Line 343 and at several other places as well – authors are listing residue numberings for different organism. It is fine but distracting as well. I suggest that authors should stick to one numbering scheme (like from human GR). If author wants, they can include a figure with sequence alignment and residue numbering so reader can refer to that figure for residue numbering in their desired organism.  

5. Figures – at several places the authors describe the hydrogen bonding/ionic interactions between different residues but in the figures, they are not showing the interactions. I suggest that the author should shows the interactions they are describing in the main text in the figures as well.

6. with so many colors Figure 11 seems very cluttered and hard to understand. Can the authors improve the figure while conveying the same message? For example – the three layers can be colored differently, or the protein can be colored as spectrum from N terminus to C terminus and/or helices can be labeled. It will also help to show the ligand as spheres as its hard to see.

7. Line 702 – it is not clear why the mutant was used. Mutations were made to stabilize the protein so that it can be crystallized. It should be mentioned here. Its not uncommon and unreasonable to use the more stable mutant proteins for crystallization. Several GPCRs were crystallized this way. Also, since these mutations have no functional significance (at-least not shown experimentally), it is not necessary and relevant to list all these mutations here. 

Author Response

Response to Reviewer 2 Comments

We cordially thank the reviewer for taking the time to review and for all the suggestions that have contributed to improving our manuscript.

Point 1: Line 207-208 – there is no reference for phosphorylation at Ser211.

Response 1: Added citation for phosphorylation at hGR Ser211:
Wang, Z.; Frederick, J.; Garabedian, M.J. Deciphering the Phosphorylation “Code” of the Glucocorticoid Receptor in Vivo * 210. Journal of Biological Chemistry 2002, 277, 26573–26580, doi:10.1074/jbc.M110530200.

Point 2:
Line 270-272 – coordination bond is not van der walls interaction.

Response 2: Deleted the statement “via Van der Waals interactions” as it is incorrect.

Point 3: Table 1 – pdb id 1RGD – method is described as “solution NMR” which is fine but then in same table, there are other structures where the method is written as “NMR” giving the impression that the other structures are solved by different method like solid state NMR which is not correct. The authors need to be consistent.

Response 3: Adapted 1GDC and 2GDA to “Solution NMR” for consistency with 1RGD.

Point 4: Line 343 and at several other places as well – authors are listing residue numberings for different organism. It is fine but distracting as well. I suggest that authors should stick to one numbering scheme (like from human GR). If author wants, they can include a figure with sequence alignment and residue numbering so reader can refer to that figure for residue numbering in their desired organism.

Response 4: Added figure with cross-species multiple sequence alignment and numbering. Deleted the residue numberings in the text, where deemed necessary, and inserted a reference to the sequence alignment figure. Residue numberings from different organisms were kept in some instances as, in our opinion, it helps readers to quickly search for a residue in the text, regardless of species.

We decided against sticking to one numbering scheme (human GR) because, if an article studied non-human GR, it feels incorrect to report those results for human GR.

Figure 1. Cross-species protein sequence alignment of GR. Comparison of GR from Homo sapiens (hGR; UniProt ID P04150), Rattus norvegicus (rGR; UniProt ID P06536) and Mus musculus (mGR; Uniprot ID P06537).

Point 5: Figures – at several places the authors describe the hydrogen bonding/ionic interactions between different residues but in the figures, they are not showing the interactions. I suggest that the author should shows the interactions they are describing in the main text in the figures as well.

Response 5: Figures were adapted, they now display relevant hydrogen bonds. Displaying additional interactions (for example VdW interactions) would quickly crowd figures and interfere with annotations.

Old figure 6

New figure 6 (changed from 6 to 7 because of the extra figure 4)

Old figure 7

New figure 7 (changed from 7 to 8 because of the extra figure 4)

Old figure 8

New figure 8 (changed from 8 to 9 because of the extra figure 4)

Old figure 9

New figure 9 (changed from 9 to 10 because of the extra figure 4)

Old figure 10

New figure 10 (Orientation was changed for better visualization of monomer 1 interactions with DNA) (changed from 10 to 11 because of the extra figure 4)

Old figure 12

New figure 12 (changed from 12 to 13 because of the extra figure 4)

Old figure 13

New figure 13 (changed from 13 to 14 because of the extra figure 4)

Point 6: with so many colors Figure 11 seems very cluttered and hard to understand. Can the authors improve the figure while conveying the same message? For example – the three layers can be colored differently, or the protein can be colored as spectrum from N terminus to C terminus and/or helices can be labeled. It will also help to show the ligand as spheres as its hard to see.

Response 6: Protein is colored as spectrum from N terminus to C terminus and helices were labeled. Dexamethasone is depicted as a sphere.

Old figure 11

New figure 11 (changed from 11 to 12 because of the extra figure 4)

Figure 2. (a) Protein sequence of the GR LBD. (b) Structural overview of the GR LBD in complex with dexamethasone and a TIF2 coactivator motif (PDB ID: 1M22) [143]. Dexamethasone is depicted as a sphere colored in red/salmon.

Point 7: Line 702 – it is not clear why the mutant was used. Mutations were made to stabilize the protein so that it can be crystallized. It should be mentioned here. Its not uncommon and unreasonable to use the more stable mutant proteins for crystallization. Several GPCRs were crystallized this way. Also, since these mutations have no functional significance (at-least not shown experimentally), it is not necessary and relevant to list all these mutations here.

Response 7: Mutations were deleted as it is indeed not relevant to list them.